# PXL1 and SERKs act as receptor–coreceptor complexes for the CLE19 peptide to regulate pollen development

Ying Yu[1,6], Wen Song [2,3,4,6], Nuo Zhai[3,6], Shiting Zhang[1], Jianzheng Wang [1], Shuangshuang Wang[1], Weijia Liu[3], Chien-Hsun Huang [1], Hong Ma [5], Jijie Chai [3,4] & Fang Chang [1] ✉

Gametophyte development in angiosperms occurs within diploid sporophytic structures and requires coordinated development; e.g., development of the male gametophyte pollen depends on the surrounding sporophytic tissue, the tapetum. The mechanisms underlying this interaction remain poorly characterized. The peptide CLAVATA3/EMBRYO SURROUNDING REGION-RELATED 19 (CLE19) plays a "braking" role in preventing the harmful over-expression of tapetum transcriptional regulators to ensure normal pollen development in *Arabidopsis*. However, the CLE19 receptor is unknown. Here, we show that CLE19 interacts directly with the PXY-LIKE1 (PXL1) ectodomain and induces PXL1 phosphorylation. PXL1 is also required for the function of CLE19 in maintaining the tapetal transcriptional regulation of pollen exine genes. Additionally, CLE19 induces the interactions of PXL1 with SOMATIC EMBRYOGENESIS RECEPTOR-LIKE KINASE (SERK) coreceptors required for pollen development. We propose that PXL1 and SERKs act as receptor and coreceptor, respectively, of the extracellular CLE19 signal, thereby regulating tapetum gene expression and pollen development.

Pollen grains are the haploid male gametophytes in angiosperms and are essential for plant fertility. Pollen development occurs in the anther locule, which is surrounded by four distinct cell layers comprising the epidermis, endothecium, middle layer, and tapetum, from outside to inside. The tapetum directly surrounds the locule, where the microspores are produced and develop into mature pollen, and is known to act as a supplier of signals, nutrients, and sporopollenin precursors for pollen development and pollen wall formation[1–7].

Previous studies have demonstrated a conserved genetically defined transcriptional pathway that regulates tapetum function and the tapetal transcriptional network essential for pollen development.

Multiple transcription factors (TFs) in this transcriptional pathway are conserved between major lineages of angiosperms (eudicots such as *Arabidopsis*, and monocots, such as rice). In *Arabidopsis*, from upstream to downstream they include: *DYSFUNCTIONAL TAPETUM1* (*DYT1*), *bHLH010/089/091*, *MYB35/DEFECTIVE IN TAPETAL DEVELOPMENT AND FUNCTION* (*TDF1*), *ABORTED MICROSPORE* (*AMS*), *MYB103/MYB80*, *MALE STERILITY1* (*MS1*), and *MYB99*[1–7]. DYT1 is considered a master regulator of the tapetum transcriptional network, as it is the earliest known male-specific regulator and is required for normal expression of more than 1000 anther genes[8–13]. *bHLH010*, *bHLH089*, and *bHLH091*, which function downstream of DYT1 and are jointly

[1]State Key Laboratory of Genetic Engineering, Ministry of Education Key Laboratory of Biodiversity Sciences and Ecological Engineering, School of Life Sciences, Fudan University, Shanghai 200438, China. [2]State Key Laboratory of Plant Environmental Resilience, College of Biological Sciences, China Agricultural University, 100193 Beijing, China. [3]Innovation Center for Structural Biology, Tsinghua-Peking Joint Center for Life Sciences, School of Life Sciences, Tsinghua University, 100084 Beijing, China. [4]Max-Planck Institute for Plant Breeding Research, Institute of Biochemistry, University of Cologne, 50829 Cologne, Germany. [5]Department of Biology, The Huck Institutes of the Life Sciences, The Pennsylvania State University, University Park 16802 PA, USA. [6]These authors contributed equally: Ying Yu, Wen Song, Nuo Zhai. ✉e-mail: fangchang@fudan.edu.cn

required for tapetum development and pollen fertility, form positive-feedback regulatory loops with DYT1 by enhancing DYT1 localization in the nucleus[8].

Given the positive-feedback regulatory loops between DYT1 and its downstream bHLH TFs, a "braking" factor/pathway is necessary to prevent harmful overexpression of the tapetum transcriptional network and maintain a normal functional level. Indeed, the peptide signal CLAVATA3/EMBRYO SURROUNDING REGION-RELATED 19 (CLE19) and some of its functionally redundant CLE family members play a "braking" role by limiting the expression of *AMS* to maintain the proper level of the tapetum transcriptional network and the formation of appropriate amount of pollen exine[14]. Specifically, in the anthers of transgenic plants expressing an antagonistic CLE19$_{G6T}$ (*DN-CLE19*) construct that was expressed under the control of the *CLE19* promoter exhibited dominant pollen developmental defects, the reduction in CLE function causes deleterious overexpression of the *AMS-MYB103/80-MS1* transcriptional cascades and downstream genes for pollen exine formation. Such overexpression subsequently causes excess accumulation of pollen exine materials covering the pollen surface and affects pollen development and viability. In contrast, overexpression of CLE peptides excessively inhibits the expression and function of *AMS* and downstream networks, thereby inducing abnormal exine formation in pollen grains[14]. However, how extracellular CLE signals are received and converted into intracellular signaling pathways to modulate tapetal transcriptional networks is not known.

In this study, we showed that CLE19 directly interacts with PXL1, which has been suggested to act synergistically with PHLOEM INTERCALATED WITH XYLEM (PXY) and PXY-LIKE2 (PXL2) in the regulation of vascular development[15]. We demonstrated that CLE19 induces PXL1 phosphorylation and the interaction between the PXL1 and SERK receptor-like protein kinases. We propose that PXL1 acts a receptor and SERKs as coreceptors of CLE19 in maintaining balanced tapetum transcriptional networks needed for normal pollen development.

## Results

### CLE19 interacts directly with PXL1 and induces its phosphorylation

As part of an effort to identify cognate receptors for peptidyl ligands in Arabidopsis, a pool of synthetic CLE small peptides (Supplementary Table 1) was mixed with purified individual extracellular LRR domains of receptor-like kinases (RLKs). The putative RLK-ligand complexes were purified by gel-filtration, and the peptides were detected by mass spectrometry, as reported previously[16]. The results indicated that the CLE19 peptide was specifically copurified with PXL1 (Fig. 1a), but not CLV3, CLE3, CLE6, and others, suggesting that PXL1 may function as a receptor for CLE19. To further test this idea in vitro, we quantified the interaction between CLE19 and PXL1$^{LRR}$ using isothermal titration calorimetry (ITC) and found that CLE19 was bound to PXL1$^{LRR}$ with a dissociation constant (Kd) of -346 nM (Fig. 1b). Furthermore, we generated *35S::PXL1-3×FLAG* transgenic plants, purified PXL1-3×FLAG proteins, carried out dot blotting assays with chemically synthesized CLE19-biotin, and found that CLE19 directly interacted with the PXL1-FLAG fusion protein but not the FLAG tag (Fig. 1c). Together, these results suggested that CLE19 and PXL1 interact directly with each other and probably function as a ligand–receptor pair.

We anticipated that if PXL1 is a receptor for CLE19, then CLE19 should promote the phosphorylation of PXL1. Therefore, we generated *pPXL1::PXL1-FLAG* transgenic plants, treated 10-day-old *pPXL1::PXL1-FLAG* transgenic seedlings with CLE19 for 1.5 h, and detected the phosphorylation of PXL1 with Phos-tag SDS-PAGE and western blotting. As predicted, a phosphorylated band appeared after CLE19 treatment, whereas only the non-phosphorylated PXL1 band was present in the untreated sample (Fig. 1d). We further immunoprecipitated the PXL1-FLAG proteins using anti-FLAG antibodies and confirmed the CLE19-induced PXL1 phosphorylation using a phospho S/T antibody

(Fig. 1e). We then questioned whether phosphorylation was specifically induced by functionally active CLE19. Therefore, we treated 10-day-old *pPXL1::PXL1-FLAG* transgenic seedlings with functionally inactive CLE19$_{G6T}$[14,17], and another CLE family member, CLV3 (Fig. 1f), and found that PXL1 phosphorylation was not induced by functionally inactive CLE19$_{G6T}$ or CLV3 (Fig. 1f). Taken together, these results demonstrate that CLE19 directly interacts with PXL1 and induces PXL1 phosphorylation, whereas the interaction with PXL1 was not detected for CLV3 and several other CLE members, strongly suggesting that PXL1 functions as a receptor for CLE19.

### *PXL1*, *PXL2*, and *PXY* together are required for pollen development

CLE19 plays an important role in pollen development, and its receptor is expected to be involved in the same functions. Therefore, two T-DNA insertional mutant alleles of *PXL1*, *pxl1-1* (SALK_001782) and *pxl1-2* (SALK_128519), were obtained and subjected to anther phenotypic analysis. The expression of full-length *PXL1* from either allele was not detected with primers flanking the T-DNA insertion sites (Supplementary Fig. 1), suggesting that both are null alleles. Then, the anther and pollen exine structures of *pxl1-1* and *pxl1-2* were examined. In comparison to those of the wild type (WT), the anthers of *pxl1-1* and *pxl1-2* were slightly smaller, and the pollen count in each anther was reduced from 461 in WT anthers down to 348 in *pxl1-1* and 355 in *pxl1-2* (Fig. 2a, d). In addition, 11.2% and 9.8% of pollen grains in *pxl1-1* and *pxl1-2*, respectively, showed pollen exine defects with parts of the pollen exine being abnormally filled, in contrast to the much lower defect rate of 3.9% in WT (Fig. 2b, c, e, f). These results suggested that PXL1 is important for normal pollen development and especially for pollen exine formation. Moreover, the above pollen developmental defects of *pxl1* mutants were similar to those of the *cle19* single mutant (Fig. 2a–f), further suggesting that PXL1 and CLE19 play similar roles in pollen development. The fact that *pxl1* pollen defects are much weaker than those of the *DN-CLE19* transgenic lines, in which 62.1% of pollen exhibited exine defects, suggests that PXL1 may have functionally redundant RLKs in the pollen developmental process.

PXL1 has been shown to be closely related to PXY and PXL2, and these three RLKs likely play redundant roles in the regulation of vascular tissue development[18]. The topology of the phylogenetic tree of homologs of these RLKs in 41 land plants also suggests that PXL1/2 and PXY were generated by two duplication events. The ancestoral genes of the PXL clade and PXY clade originated due to a duplication before the divergence of monilophytes and seed plants. After that, the ancestors of PXL1 and PXL2 genes and their respective close homologs in these two clades were generated by a more recent duplication in early eudicots (Supplementary Fig. 2). In addition, PXL1 and PXL2 show 50.54.% sequence identity at the amino acid level, whereas the sequence identity between PXY and PXL1 and that between PXY and PXL2 are 29.25% and 27.44%, respectively, suggesting that the functions of PXL1 and PXL2 are more closely related.

In addition, observations in transgenic plants with the GUS reporter gene expression driven by native promoters showed that PXL1, PXL2 and PXY were all relatively highly expressed in anthers at stages 7–9, and in both the tapetum layer and reproductive cells (Fig. 2g); and analyses in both *Nicotiana benthamiana* leaf cells and protoplasts with EYFP tagged coding sequences (CDS-EYFP) driven by the 35S promoter showed that these proteins were all distributed on the PM (Fig. 2h and Supplementary Fig. 3), without detectable signals in other subcellular compartments. These results further supported their redundant functions as PM-localized receptor kinases in anther and pollen development.

To further test this idea, we obtained knockout mutants of the *PXL2* and *PXY* genes (Supplementary Figs. 4 and 5), and observed the pollen developmental phenotypes of *pxl2-1* and *pxy-3* single mutants, the *pxl1 pxl2* double mutant, and the *pxl1-1 pxl2-1 pxy-3* triple mutant.

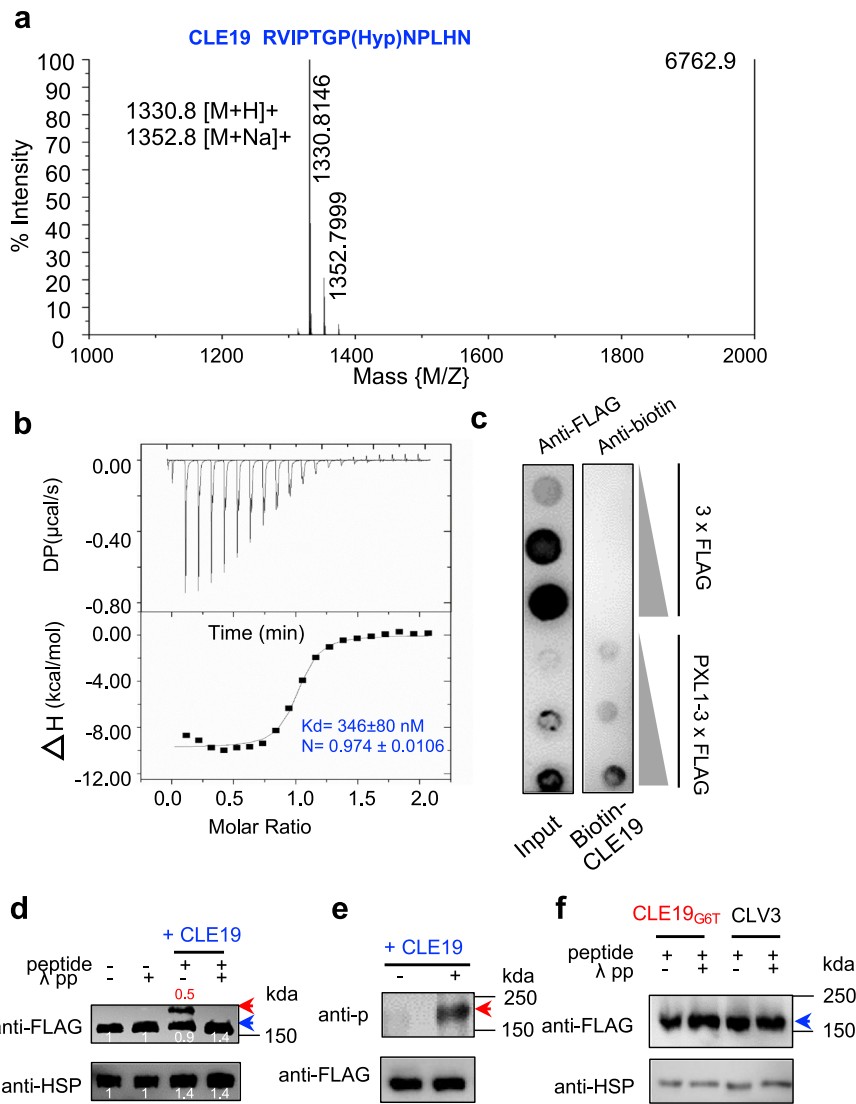

**Fig. 1 | CLE19 interacts directly with PXL1 and induces PXL1 phosphorylation.** **a** Identification of CLE19–PXL1 interaction by gel filtration-MALDI-TOF. **b** Measurement of binding affinity between CLE19 and PXL1$^{LRR}$ by ITC. Top panel: 20 injections of CLE19 solution were titrated into PXL1$^{LRR}$ solution in the ITC cell. The area of each injection peak corresponds to the total heat released by that injection. Bottom panel: the binding isoform for CLE19 and PXL1$^{LRR}$ interaction. The integrated heat is plotted against the molar ratio of CLE19 to PXL1$^{LRR}$. Data fitting revealed a binding affinity of ~346 nM. **c** Dot blot showing the direct interaction between PXL1 and CLE19. The left panel shows increasing concentrations of immobilized 3xFLAG peptide and PXL1-3xFLAG fusion protein on nitrocellulose filter membrane. The right panel shows that PXL1-3xFLAG but not 3xFLAG can bind to CLE19. **d**–**f** CLE19 specifically induces the phosphorylation of PXL1 in vivo. **d** Seedlings of *pPXL1::PXL1-FLAG* transgenic plants were subjected to CLE19 treatment or left untreated. Electrophoretic mobility of the phosphorylated PXL1-FLAG band (indicated by a red arrowhead) was altered in the phos-tag gel after a 1.5-h incubation with CLE19, which was abolished by treated with λpp. HSP was used to indicate the input amount. The intensity of each band in (**d**) was measured using ImageJ software. After measurement, the intensity of the non-phosphorylated PXL1-FLAG band in the first lane from left to right was set as a reference value of 1, and the value of each detected band in the second, third and fourth lanes were expressed as the ratio of the detected bands to that of the reference band. Similarly, the intensity of HSP in the first lane from left to right was set to 1, and the intensity of other bands was presented as the ratio to this HSP band. **e** Seedlings of *pPXL1::PXL1-FLAG* transgenic plants were treated with 20 μm CLE19 for 1.5 h, and then PXL1-FLAG proteins were pulled down by FLAG beads. The CLE19-induced phosphorylation of PXL1 was verified with a pT/S antibody, and the anti-FLAG antibody was used to indicate the input amount. **f** Neither CLE19$_{G6T}$ nor CLV3 induced the phosphorylation-related migration of PXL1. In (**d**–**f**), the phosphorylated bands are indicated by red arrows, and the unphosphorylated bands are indicated with blue arrowheads. Three times experiments were repeated with similar results for (**d**–**f**).

Anthers from *pxl2-1* and *pxy-3* single mutants were slightly smaller, and pollen counts in each anther were also reduced in these mutants, similar to the case in *pxl1-1* and *pxl1-2* (Fig. 2a–d). In addition, similar to the phenotypes of the *pxl1* mutants, a fraction of pollen from *pxl2-1* and *pxy-3* was defective, with excess pollen wall materials (Fig. 2b, c, f). The *pxl1-1 pxy-3* and *pxl1-2 pxl2-1* double mutants showed smaller anther sizes and aborted pollen grains, and the *pxl1-1 pxl2-1 pxy-3* triple mutant exhibited pollen wall defects that were more severe than those of the single and double mutants (Fig. 2a–f). These results strongly

suggested that *PXL1*, *PXL2*, and *PXY* are functionally redundant in the regulation of pollen development. To semi-quantitatively describe the severity of pollen exine defects in various genotypes, we classified the defects into two groups (Fig. 2e). The pollen grains with normal size but with less than half of pollen exine area being abnormally filled were defined as having moderate defects (moderate-D for short), whereas those with a collapsed morphology and most of the pollen exine abnormally covered were defined as having severe defects (severe-D for short). Interestingly, only the *pxl1* single

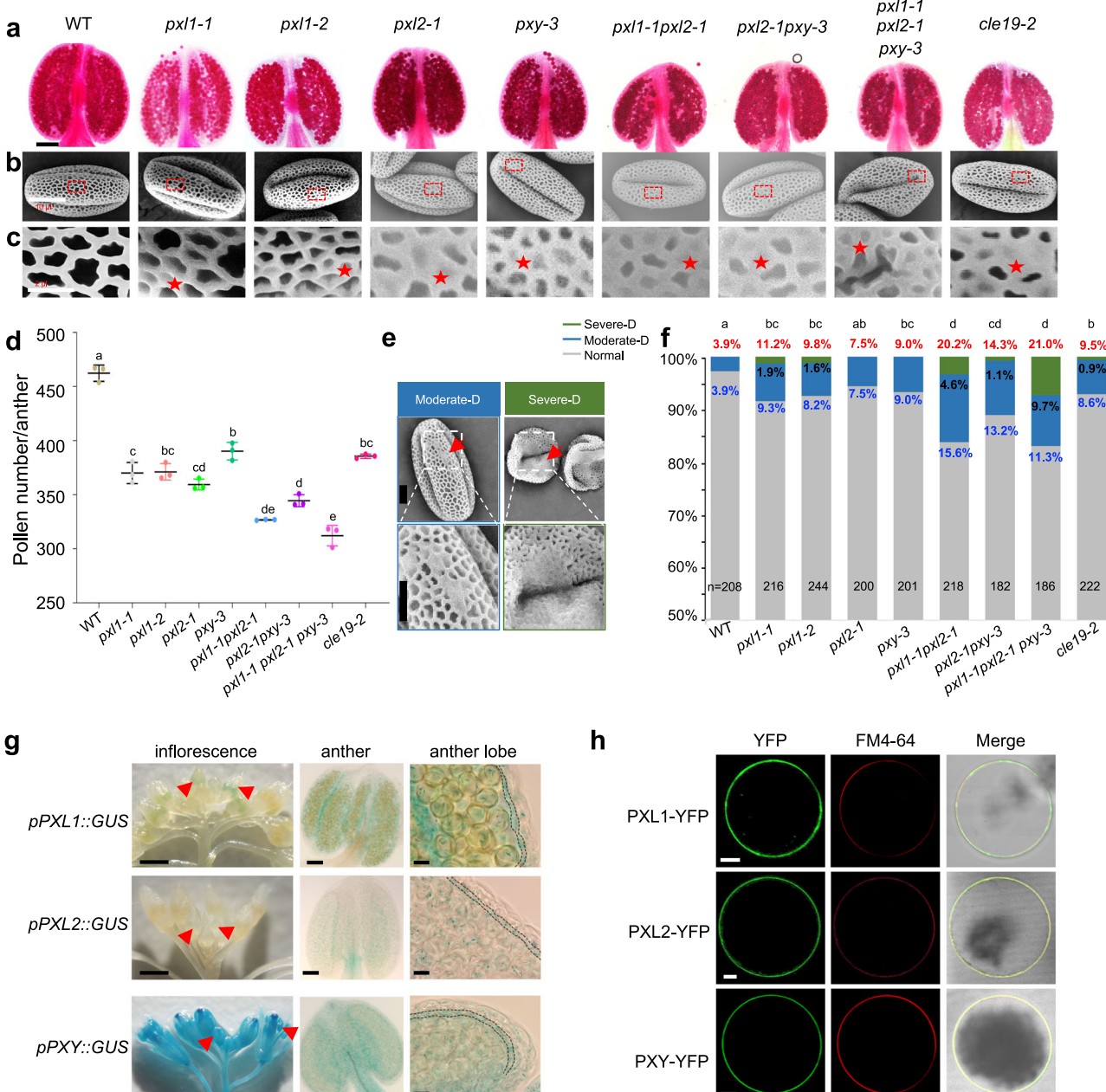

**Fig. 2 | *PXL1*, *PXL2* and *PXY* together are required for pollen development.**
**a–c** Phenotypic observations of anther and pollen of WT, *pxl1*, *pxl2*, *pxy* single and *pxl1 pxl2*, *pxl2 pxy* double and *pxl1 pxl2 pxy* triple mutants. **a** Alexander-stained anthers and **b**, **c** SEM images of pollen and pollen exines are shown. Bar = 100 μm for (**a**), 10 μm for (**b**), and 2 μm for (**c**). The red stars indicate the pollen exine defects. **d** Quantification of pollen amounts per anther in the indicated genotypes. Three independent biological replicates were performed, and ten anthers were used for analysis for each replicate. The data are shown as the mean ± SD, each dot shows the average value for one biological replicate. Different letters represent significant difference between each other, $p < 0.05$, one-way ANOVA with Tukey multiple comparison test. Exact $p$ values are 2.04e$^{-4}$ for *pxl1-1* vs. WT, 1.21e$^{-4}$ for *pxl1-2* vs. WT, 3.89e$^{-5}$ for *pxl2-1* vs. WT, 3.46e$^{-4}$ for *pxy-3* vs. WT, 6.28e$^{-6}$ for *pxl1-1pxl2-1* vs. WT, 2.56e$^{-5}$ for *pxl2-1pxy-3* vs. WT, 2.79e$^{-5}$ for *pxl1-1pxl2-1 pxy-3* vs. WT, 6.82e$^{-5}$ for *cle19-2* vs. WT. **e** SEM images showing the two types of defective pollen exines, which we defined as moderate-D and severe-D. Bar = 5 μm for the upper pictures, bar = 4 μm for the bottom enlarged pictures. **f** Proportion of pollen with

normal (gray), moderate-D (blue) and severe-D (green) exine defects from various genotypes. The statistical tests were performed between normal and the sum of moderate-D and severe-D. Letters were assayed based on calculation by chi-square, and *n* indicates the number of anthers counted for each genotype. Exact *p* values are 0.0053 for *pxl1-1* vs. WT, 0.016 for *pxl1-2* vs. WT, 0.134 for *pxl2-1* vs. WT, 0.0422 for *pxy-3* vs. WT, 2.62e$^{-7}$ for *pxl1-1pxl2-1* vs. WT, 0.0003 for *pxl2-1pxy-3* vs. WT, 1.65e$^{-7}$ for *pxl1-1pxl2-1pxy-3* vs. WT, 0.0216 for *cle19-2* vs. WT. **g** Expression of *PXL1*, *PXL2* and *PXY* genes in inflorescences, anthers and anther lobes examined using *PXL1::GUS*, *PXL2::GUS* and *PXY::GUS* reporter lines. Bar = 100 μm for inflorescences, 50 μm for anthers, 10 μm for anther lobes. The expression in early floral buds is indicated by the red arrowhead, and the anther tapetum layer in each lobe is indicated by the dotted lines. **h** Subcellular localization of PXL1-YFP, PXL2-YFP and PXY-YFP were observed in a transient expression system of *Nicotiana benthamiana* protoplasts. YFP signal is shown in green, and FM4-64 staining is shown in red. Bar = 5 μm. Three times experiments were repeated with similar results for (**g**, **h**).

mutants, the *pxl1 pxl2* double mutant, and the *pxl1 pxl2 pxy* triple mutants (i.e., plants with the *pxl1* mutation) produced pollen grains with a severe-D phenotype, whereas the *pxl2* and *pxy* single mutants and the *pxl2 pxy* double mutant exhibited a moderate-D phenotype (Fig. 2f), suggesting that *PXL1* likely plays the major role among these three partially redundant RLK genes.

### *DN-PXL1* presented the same pollen developmental defects as *DN-CLE19*

Deletion of the cytoplasmic kinase domain of RLKs has been shown to have a dominant negative (DN) effect on the WT receptor copy[19,20]. We reasoned that the overexpression of a DN form of PXL1 could inhibit the function of redundant RLKs by competing for the ligand; therefore, we generated DN-PXL1 by deleting its kinase domain (Supplementary Fig. 5a) and obtained *pPXL1::DN-PXL1-4xMYC* (*DM* for short) and *pPXL1::DN-PXL1-EYFP* (*DE* for short) transgenic plants. Two *DM* transgenic lines, *DM#8* and *DM#25*, and two *DE* transgenic lines, *DE#12* and *DE#20*, were identified with an ~3:1 segregation ratio for hygromycin resistance to sensitivity, which is consistent with a single locus of the T-DNA insertion. The relative expression of *DN-PXL1* transcripts in these four lines was estimated using qRT–PCR and quantified according to the relative expression ratio of *PXL1-LRR* (detected with primers targeting the LRR region) to that of *FL-PXL1* (detected with primers targeting the kinase domain) (Supplementary Fig. 5). The relative expression ratios in *DM#8*, *DM#25*, *DE#12* and *DE#20* plants were 5.91-, 7.14-, 2.98-, 1.63-fold that in the WT, respectively (Supplementary Fig. 5).

We then observed the pollen development phenotypes of these four *DN-PXL1* lines and found that they all showed reduced anther size and pollen counts (Fig. 3a–f). Compared to that of the WT, which produced an average of ~460 pollen grains per anther, the average pollen counts of *DN-PXL1#8*, *DN-PXL1#25*, *DN-PXL1#12* and *DN-PXL1#20* anthers were severely reduced, to 200, 183, 267 and 236, respectively (Fig. 3f). In addition, 42.8, 60, 53.8, and 43.4% of pollen grains in these four *DN-PXL1* plants showed more extensive pollen exine surface defects, which is similar to the case in *DN-CLE19*, which contained 62.1% defective pollen grains (Fig. 3g). Interestingly, 35.7%, 36.4%, 49.2% and 38.6% pollen from *DM#8*, *DM#25*, *DE#12* and *DE#20* showed severe-D defects, and 7.1%, 23.6%, 4.6% and 4.8% showed moderate-D defects, respectively (Fig. 3g), which is similar to the profile observed for *DN-CLE19*, which contained 48.27% severe-D and 13.8% moderate-D pollen grains (Fig. 3g). For the two MYC tag transgenic lines (*DM#8* and *DM#25*), the expression level of *DN-PXL1* in *DM#25* was higher than that in *DM#8*. Consistently, the pollen exine defects of *DM#25* were statistically more severe than those of *DM#8*. Similarly, for the two EYFP tag transgenic lines (*DE#12* and *#20*), the expression level of *DN-PXL1* was also correlated with the severity of the pollen exine defects. Thus, the severity of the pollen exine defects is correlated with the expression level of *DN-PXL1*, when the two MYC-tagged transgenic lines are considered separately from the two YFP-tagged transgenic lines. Together, these results strongly support the idea that PXL1 and redundant RLKs act in the same functional module as CLE19 in the regulation of pollen exine formation.

As the *DN-PXL1* transgenic lines were all in the WT background and still possessed two copies of WT *PXL1*, we crossed the *DM#8 and DE#12* lines with the *pxl1-1* mutant and generated *DM#8/pxl1-1+/−* and *DE#12/pxl1-1−/−* plants. In these two lines, the pollen count per anther was reduced to 159 and 80, and the proportion of pollen with severe-D exine defects was increased to 45.2% and 60.7%, respectively (Fig. 3e–g). These data further supported the important role of PXL1 and its redundant RLKs in the regulation of pollen development.

We further tested the expression of key genes encoding transcription factors (TFs) and enzymes for pollen exine formation in *DM#25* and *DE#12* inflorescences. The expression of *AMS*, *MYB103/MS188*, and *MS1* was increased significantly in the *DN-PXL1* transgenic

lines, but the expression of *DYT1* was not changed (Fig. 3h). In addition, the expression of genes involved in pollen exine formation, including *ACOS5*, *CYP98A8*, *CYP86C3*, *At5g55320*, *UGT72E2*, *PAL4*, and *At1g76470* was enhanced (Fig. 2i). The changes in expression of these genes are similar in *DN-PXL1* and *DN-CLE19*, as previously reported[14,17], further suggesting that PXL1 and CLE19 act in the same signaling pathway in pollen development.

### PXL1 is required for the function of CLE19

Then, we tested whether PXL1 acts as a receptor downstream of CLE19. We reasoned that if this were the case, then DN-PXL1 would suppress the pollen developmental defects caused by CLE19 overexpression (CLE19-OX). Therefore, we generated *CLE19-OX/DN-PXL1* double transgenic plants by transforming the *35S::CLE19-FLAG* construct into *DM#8* transgenic plants, as well as into Columbia WT as a control.

The expression level of *CLE19* in each line was measured by qRT–PCR, and three new homozygous CLE19-OX lines (*C#11*, *C#16* and *C#18*) and two homozygous *CLE19-OX/DN-PXL1* double transgenic lines (*CDM#3* and *CDM#4*) were chosen for further phenotypic analyses. In comparison to the expression in the WT, the expression of *CLE19* in these three *CLE19-OX* homozygous transgenic lines was increased to nearly 2000-, 400-, and 600-fold, and that in the two *CLE19-OX/DN-PXL1* homozygous lines was enhanced to ~1200- and 900-fold (Fig. 4a), indicating strong overexpression of *CLE19* in these lines.

Then, the pollen development phenotypes of these *CLE19-OX* and *CLE19-OX/DN-PXL1* lines were analyzed. Anthers from the three *CLE19-OX* lines and the two *CLE19-OX/DN-PXL1* lines were all smaller and had much less pollen in each anther than did WT anthers (Fig. 4b). For further quantification, in addition to the moderate-D and severe-D (Fig. 2e), we defined two further classes of pollen exine defects: (1) severe-C, severe CLE19-OX defects with collapsed pollen grains and a large portion of disconnected pollen exine, and (2) moderate-C, moderate CLE19-OX defects with nearly normal pollen morphology but a portion of disconnected pollen exine (Fig. 4c).

In WT anthers, 97% of the pollen grains showed a well-organized pollen exine structure, and only 3% exhibited moderate-D exine defects (Fig. 4d). In comparison, only 68 and 60% of the pollen in *C#11* and *C#18* plants was normal, and 32 and 30% showed pollen wall defects. Specifically, 26% and 17% of pollen showed severe-C defects, and 6% and 24% showed moderate-C defects, in *C#11* and C#18 plants, respectively (Fig. 4d–f). Interestingly, in *CDM#3* and *CDM#4* double transgenic plants, the proportion of pollen with CLE19-OX-like defects was dramatically reduced to 18% and 20%, with 9% and 3% pollen showing severe-C defects and 9% and 17% showing moderate-C defects, respectively (Fig. 4d, e). Moreover, 43% (17% + 26%) and 39% (29% + 10%) of the total pollen showed *DN-PXL1*-like exine defects, close to the percentages in *pPXL1::DN-PXL1* transgenic plants (Fig. 4d, f).

CLE19 is known as a negative regulator of the pollen developmental pathway, and the expression of pollen exine formation-related enzyme genes was severely suppressed by CLE19 overexpression. Therefore, we further tested the expression of these genes in *CLE19-OX/DN-PXL1* (*CDM*) double transgenic plants. Consistently, the expression of genes encoding pollen exine-related enzymes was increased in *CLE19-OX/DN-PXL1* (*CDM*) double transgenic plants (Fig. 4j). As such, *DN-PXL1* suppressed the reduction of pollen gene expression due to *CLE19*-OX and *CLE19-OX*-like pollen exine defects, such that the pollen exine defects and pollen gene expression of *CLE19-OX/DN-PXL1* were similar to those of *DN-PXL1*, strongly suggesting that *DN-PXL1* is epistatic to *CLE19-OX* and thus indicating that the CLE19 signal and CLE19 signal through PXL1 are (at least partially) dependent on the presence of a functional PXL1 receptor. This conclusion is also strongly supported by the findings that CLE19 binds to PXL1 (Fig. 1a–c), promotes phosphorylation of PXL1 (Fig. 1d–f), and induces the interaction of the PXL1 receptor and SERK coreceptors (see below).

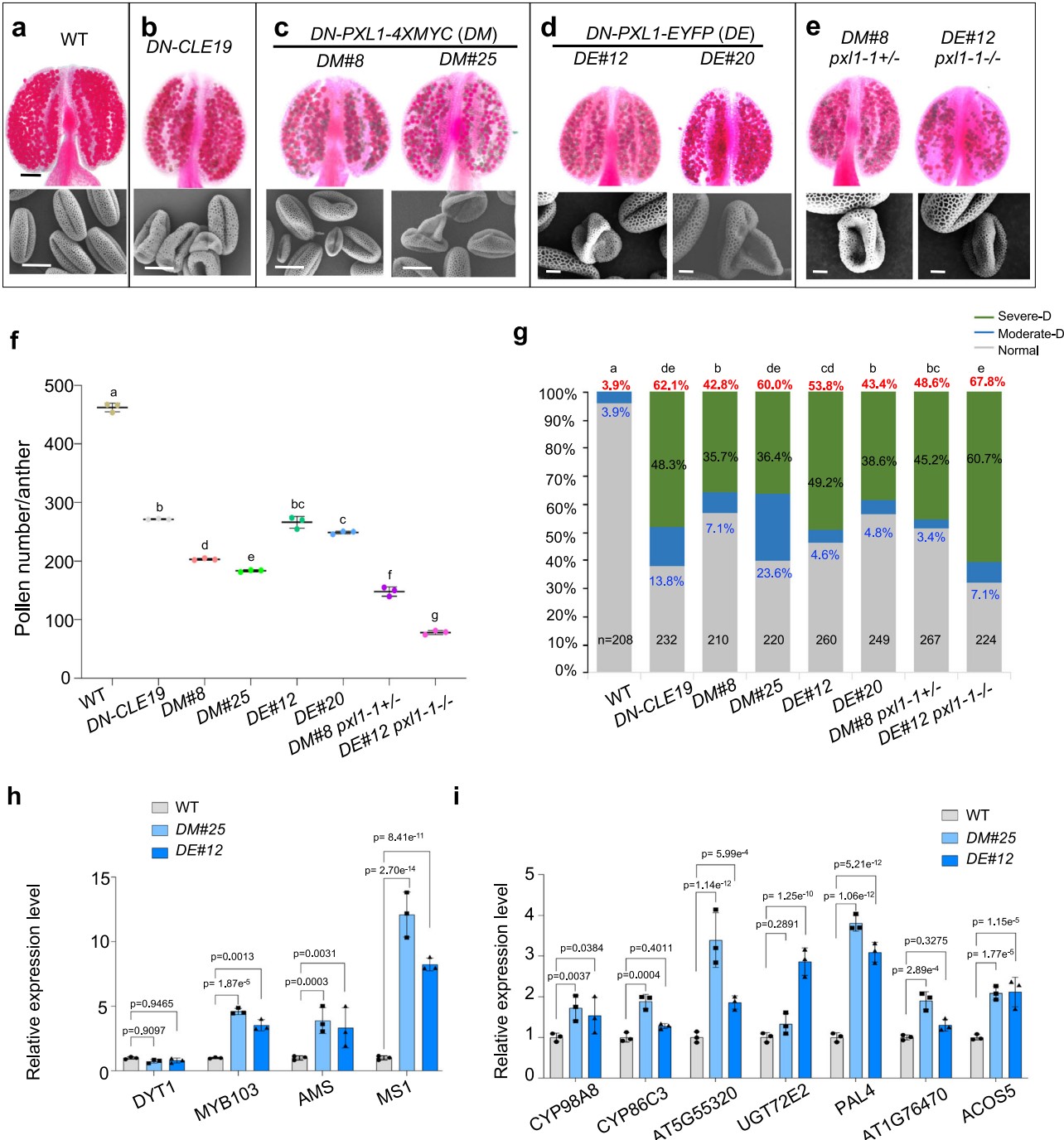

**Fig. 3 | PXL1 is required for normal pollen exine formation. a–e** Phenotypic analyses of anther and pollen grains of the WT, *DN-CLE19*, *DM#8*, *DM#25*, *DE#12*, *DE#20*, *DM#8 pxl1+/–*, *DE#12 pxl1-1-/–* transgenic plants. Alexander-stained anthers and SEM images of pollen are shown. Bar = 100 μm for the anthers and 10 μm for the SEM images. **f** Quantification of pollen amounts per anther in the indicated genotypes. Three independent biological replicates were performed, and ten anthers were used for analysis for each replicate. The data are shown as the mean ± SD, each dot shows the average value for one biological replicate. Different letters represent significant difference between each other, *p* < 0.05, one-way ANOVA with Tukey multiple comparison test. Exact *p* values are 1.66e$^{-6}$ for *DN-CLE19* vs. WT, 5.13e$^{-7}$ for *DM#8* vs. WT, 3.94e$^{-7}$ for *DM#25* vs. WT, 1.15e$^{-5}$ for *DE#12* vs. WT, 1.26e$^{-6}$ for *DE#20* vs. WT, 9.53e$^{-7}$ for *DM#8 pxl1-1+/–* vs. WT, 1.44e$^{-7}$ for *DE#12*

*pxl1-1-/–* vs. WT. **g** Proportion of pollen with normal, moderate-D and severe-D exine defects of various genotypes. The statistical tests were performed between normal and the sum of moderate-D and severe-D. Letters were assayed based on calculation by chi-square. *N* indicates the number of anthers counted for each genotype. Exact *p* values are 6.91e$^{-61}$ for *DN-CLE19* vs. WT, 1.10e$^{-29}$ for *DM#8* vs. WT, 2.79e$^{-58}$ for *DM#25* vs. WT, 1.47e$^{-40}$ for *DE#12* vs. WT, 2.49e$^{-27}$ for *DE#20* vs. WT, 1.35e$^{-32}$ for *DM#8 pxl1-1+/–* vs. WT, 3.83e$^{-81}$ for *DE#12 pxl1-1-/–* vs. WT. The relative expression of *DYT1*, *AMS*, *MYB103*, *MS1* (**h**) and of *CYP98A8*, *CYP86C3*, AT5G55320, *UGT72E2*, *PAL4*, AT1G76470 and *ACOS5* (**i**) in WT, *DM#25* and *DE#12*, as evaluated by qRT-PCR. *ACTIN* was used as the internal control. Three biological replicates were performed. Each dot shows the result for one biological replicate. Data are shown as the mean ± SD. *p* values were calculated by Student's *t* test, two sided for (**h**, **i**).

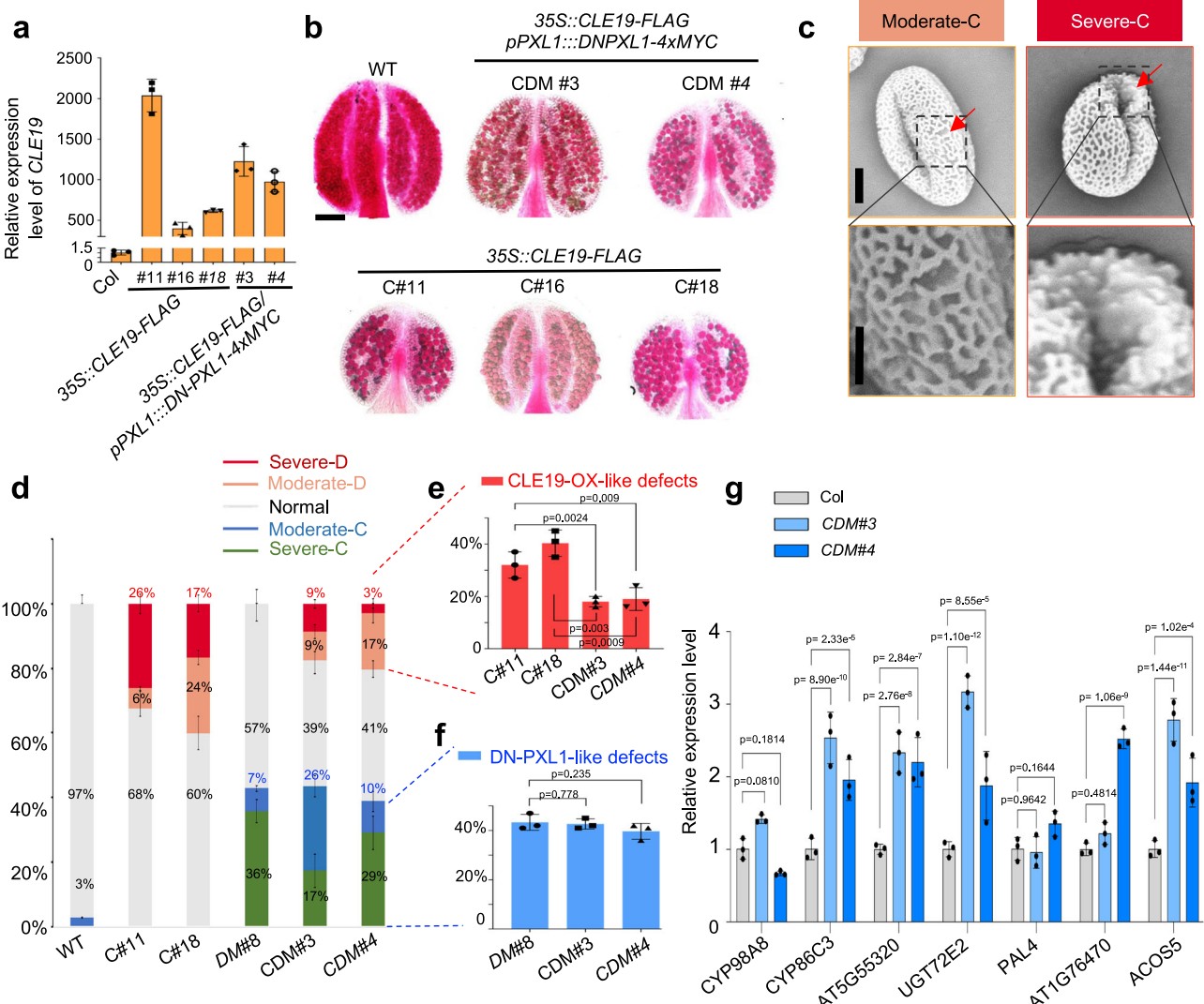

**Fig. 4 | DN-PXL1 strongly suppressed the pollen wall defects caused by CLE19-OX. a** Relative expression of *CLE19* in the inflorescences of various *35S::CLE19-FLAG and 35S::CLE19-FLAG/pPXL1::DN-PXL1* transgenic plants. The bars indicate the standard error of the mean (SEM) of three replicates. **b** Alexander-staining results of anthers from WT, *35S::CLE19-FLAG #11(C#11), 35S::CLE19-FLAG #16 (C#16), 35S::CLE19-FLAG #18 (C#18), 35S::CLE19-FLAG/pPXL1::DN-PXL1#3 (CDM#3), 35S::CLE19-FLAG/pPXL1::DN-PXL1#4 (CDM#3)* plants. Bar = 100 μm. **c** SEM showing the two types of defective pollen exines in the *35S::CLE19-FLAG and 35S::CLE19-FLAG/pPXL1::DN-PXL1* transgenic plants, which we defined as moderate-C and severe-C, respectively. Bar = 6 μm for the upper images, and 3 μm for the lower (enlarged) images. Three times experiments were repeated with similar results.

**d** Statistical analyses of the proportions of the five types of pollen grains in the WT, *35S::CLE19-FLAG, pPXL1::DN-PXL1 and 35S::CLE19-FLAG/pPXL1::DN-PXL1* transgenic anthers. **e** Proportion of CLE19-OX-like pollen grains in C#11, C#18, CDM#3, and CDM#4 in (**d**). **f** Proportion of DN-PXL1-like pollen in (**d**). The data in (**d–f**) are shown as the mean ± SD of three biological replicates. Each dot showed the result for one biological replicate. *p* values were calculated by *t*-test, two sided. **g** Relative expression of *CYP98A8*, *CYP86C3*, AT5G55320, *UGT72E2*, *PAL4*, AT1G76470 and *ACOS5* in the WT and the two *35S::CLE19-FLAG/pPXL1::DN-PXL1* transgenic plants. Three biological replicates were performed. Each dot showed the result for one biological replicate. Data are shown as the mean ± SD. *p* values were calculated by Student's *t* test, two sided.

## SERKs serve as coreceptor of PXL1 in the regulation of pollen development

SOMATIC EMBRYOGENESIS RECEPTOR-LIKE KINASES (SERKs) act as coreceptors in multiple RLK-mediated signaling pathways, including the TDIF-PXY pathway, in which PXL1 and PXL2 are suggested to act synergistically with PXY in regulating vascular-tissue development in the stem[15], and the TPD-EMS/EXS pathway in the regulation of tapetum differentiation[21,22]. Therefore, we tested whether SERKs also function as coreceptors of PXL1 in the regulation of pollen wall formation.

We first analyzed the pollen exine phenotypes of *SERK* gene-related mutants. As the *serk1 serk2* double homozygous mutant produced no pollen due to the absence of tapetum[23], we analyzed the phenotypes of the *serk1, serk2,* and *bak1 (serk3)* single mutants and *serk1+/− serk2 bak1* triple mutant (with one normal allele) anthers.

The *serk1, serk2* and *bak1* single mutants were normal (Supplementary Fig. 6a, b), but *serk1+/− serk2−/−bak1−/−* showed reduced anther size and pollen count (Fig. 5a), with only ~120 pollen grains in each anther (Supplementary Fig. 6b), probably because of that the dysfunction of half of the *SERK1* proteins, together with *SERK2*, partially affected the development of the tapetum and microspores and finally reduced the yield of pollen grains. Therefore, we focused our phenotypic observation on the pollen exine. The pollen exine structure of most (>90%) of the *serk2* and *bak1* pollen grains was normal, whereas the proportion of normal grains decreases to 74.3% in the *serk1+/− serk2 bak1* triple mutant, with 2.6% showing severe-D defects and 23.1% showing moderate-D defects (Supplementary Fig. 6c). At the same time, we evaluated the relative expression level of genes involved tapetum development and pollen wall formation in

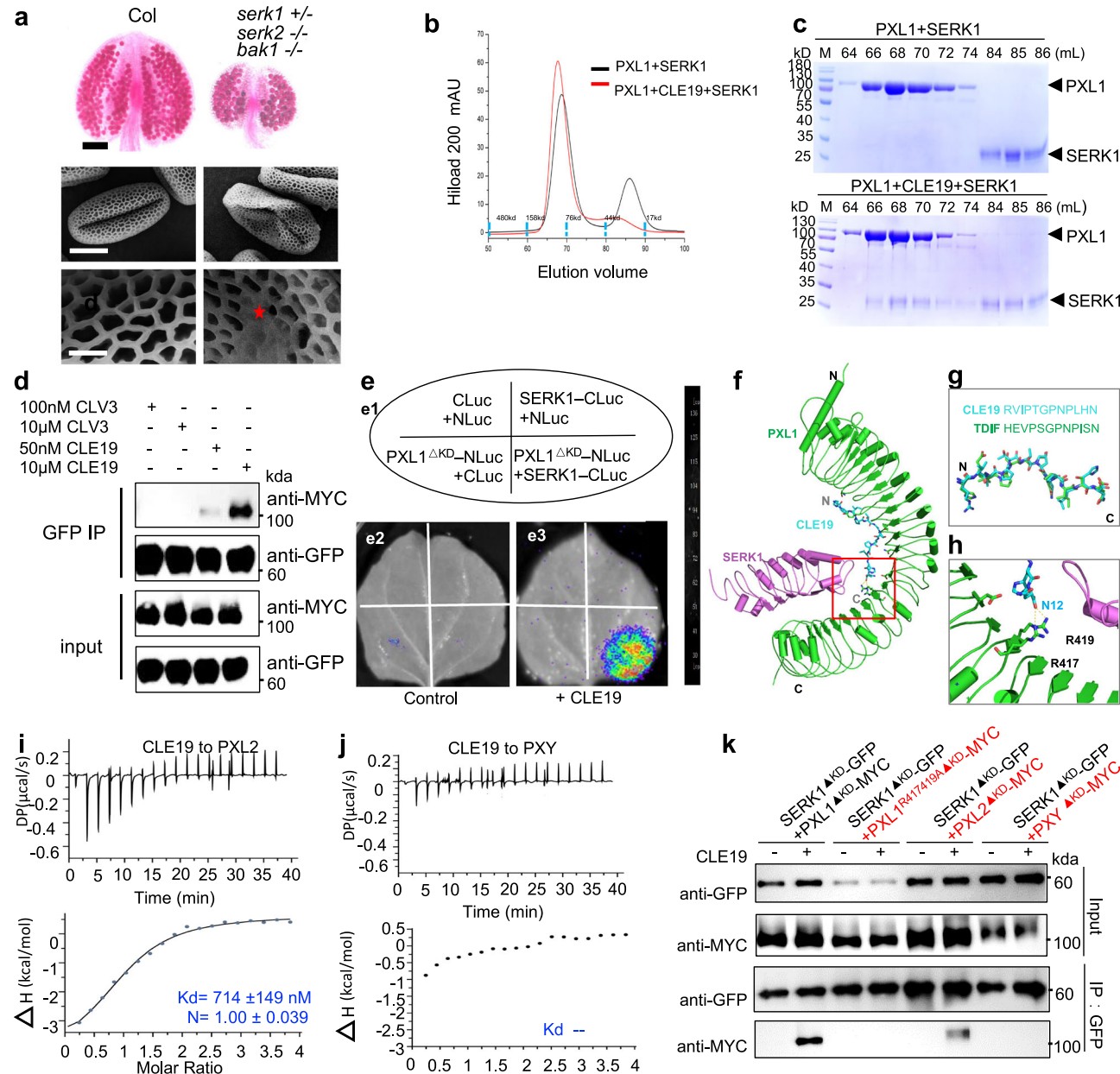

**Fig. 5 | SERKs function as coreceptors for PXL1 in pollen development.**
**a** Phenotypic analyses of Col and *serk1+/−serk2−/−bak1−/−*. Top panel, Alexander staining of anthers; middle panel, SEM observation of pollen grains; bottom panel, enlarged pollen wall structure. The red star indicates abnormal pollen exine. Bar = 100 μm for the top panel, 10 μm for the middle panel, and 2 μm for the bottom panel. Three times experiments were repeated with similar results. **b** Superposition of the gel filtration chromatograms of the PXL1LRR + SERK1LRR and PXL1LRR + CLE19 + SERK1LRR proteins. The vertical and horizontal axes represent UV absorbance (280 nm) and elution volume (ml), respectively. **c** Coomassie blue staining of the peak fractions shown in (**b**) following SDS-PAGE. M, molecular weight ladder (kDa). Three times experiments were repeated with similar results. **d** In vivo coimmunoprecipitation assay showing the PXL1ΔKD–SERK1ΔKD interaction in the presence or absence of CLE19 treatment. PXL1ΔKD, PXL1 with kinase domain deletion; SERK1ΔKD, SERK1 with kinase domain deletion. MYC-tagged PXL1ΔKD and

GFP-tagged SERK1ΔKD were coexpressed in *Nicotiana benthamiana*. CLV3 was used as a control. Three times experiments were repeated with similar results. **e** Split luciferase assays show the interaction between PXL1ΔKD and SERK1 in the presence or absence of CLE19 treatment in *Nicotiana benthamiana*; e1 shows the infiltration of the protein pairs, and e2, e3 show the protein interactions in the present of a control treatment and 20 μm CLE19, respectively. **f** The overeall modeling structure of the CLE19–PXL1LRR/SERK1LRR complex. **g** Sequence and structure comparison between CLE19 and TDIF. **h** Detailed interactions of the boxed region in (**f**). The side chain of the terminal CLE19 residue and the R417/R419 sites on the PXL1 surface are labeled. Measurements of binding affinity between CLE19 and PXL2 (**i**), and between CLE19 and PXY (**j**) by ITC. **k** In vivo co-IP assay using the *Nicotiana benthamiana* leaf transient expression system showing the interaction between PXL1ΔKD/PXL1ΔKD-R417419A/PXL2ΔKD/PXYΔKD and SERK1ΔKD in the presence or absence of CLE19 treatment. Three times experiments were repeated with similar results.

the *serk1+/− serk2−/− bak1−/−* triple mutant, and the results showed that *AMS*, *MYB103*, and *MS1* were all upregulated (Supplementary Fig. 6d), similar to the case in *DN-CLE19* or *DN-PXL1* transgenic plants. At the same time, the expression of several genes related pollen wall formation, including *CYP98A8*, *CYP86C3*, and *ACOS5*, was

upregulated in triple mutants (Supplementary Fig. 6e). These data support the idea that SERKs are required for the normal development of pollen exine.

We then tested whether PXL1 and SERKs physically form a receptor complex. We purified the extracellular LRR domains of SERK1

and tested its interaction with PXL1 through a gel filtration assay. The results showed that PXL1$^{LRR}$ could form a stable complex with SERK1$^{LRR}$ in the presence but not in the absence of CLE19 (Fig. 5b, c), indicating that the formation of PXL1-SERK1 complexes is dependent on CLE19. In addition, CLE19-dependent PXL1-SERK1 interactions were also verified with a coimmunoprecipitation (co-IP) assay through transient coexpression of the MYC-tagged truncated form of PXL1 with kinase domain deletion (PXL1$^{\Delta KD}$-MYC) and GFP-tagged SERK1 (SERK1$^{\Delta KD}$-GFP) in *Nicotiana benthamiana* and a subsequent co-IP assay with or without CLE19 treatment. The interaction was detected between PXL1$^{\Delta KD}$ and SERK1$^{\Delta KD}$ in the presence of CLE19, and the interaction signal increased with increasing CLE19 peptide concentration; however, no such interaction was detected in the absence of CLE19 or the presence of CLV3 (Fig. 5d). The interaction between PXL1$^{\Delta KD}$ and SERK1$^{\Delta KD}$ was also confirmed by a firefly luciferase complementation imaging (LCI) assay (Fig. 5e). Consistently, the interaction was detected only in the presence of CLE19. These results demonstrated that SERK1 serves as a coreceptor of PXL1 and that CLE19 promotes the interaction between PXL1 and its SERK1 coreceptor in the regulation of pollen development and pollen exine formation.

We then questioned whether other SERK members could also act as coreceptors of PXL1 to perceive the CLE19 signal. Thus, we investigated the interaction between PXL1 and BAK1/SERK3 using the same co-IP and LCI systems, and found that the PXL1-BAK1 interaction was also induced by CLE19, but not CLV3 (Supplementary Fig. 7), suggesting that BAK1 also acts as a coreceptor. In addition, CLE19 was found to promote interaction between PXL1 and SERK2, and between PXL1 and BAK1, using co-IP analysis (Supplementary Fig. 8). Together, these interactions and the phenotypes of the *serk* mutants indicate that SERK1, SERK2, and SERK3/BAK1 are redundantly required in the regulation of pollen wall formation by acting as coreceptors of PXL1 for perceiving CLE19.

According to the structure modeling results (Fig. 5f–h), the R417 and R419 sites of PXL1 are predicted to be in the PXL1-CLE19 interaction surface. Therefore, we cotransformed the 35S:PXL1$^{R417AR419A\Delta KD}$-MYC and 35S:SERK1$^{\Delta KD}$-GFP constructs into *Nicotiana benthamiana* leaves and investigated the interaction between the PXL1$^{R417AR419A\Delta KD}$ mutant protein and its coreceptors SERK1$^{\Delta KD}$, SERK2$^{\Delta KD}$, and SERK3$^{\Delta KD}$ using co-IP analysis. As predicted, the R417AR419A mutation abolished the interaction of PXL1 and its coreceptors (Fig. 5k and Supplementary Fig. 8a, b).

As PXL2 and PXY are redundantly required for pollen wall formation (Figs. 2a–f and 3), we further questioned (1) whether PXL2 and PXY physically interact with CLE19 as PXL1 does, and (2) whether CLE19 induces the interaction of PXL2 and SERKs, as well as that of PXY and SERKs. Interestingly, using ITC analysis, we found that CLE19 bound to PXL2 with a dissociation constant (Kd) of 714 nM (Fig. 5i), suggesting that the interaction of CLE19 and PXL2 is slightly weaker than that of CLE19 and PXL1. In comparison, the CLE19-PXY pair showed no interaction (Fig. 5j). Moreover, using co-IP analyses in a *Nicotiana benthamiana* leaf transient expression system, we found that CLE19 induced interactions between the PXL2 receptor and SERK1/SERK3 coreceptors but not interactions between PXY and any SERK protein (Fig. 5k and Supplementary Fig. 8). These results strongly suggested that although the PXL1/PXL2/PXY receptors and the SERK1/SERK2/SERK3 coreceptors exhibited functional redundancy, they may have different roles in serving as receptor–coreceptor complexes in mediating CLE19 signaling.

## Discussion

The cell–cell communication mediated by extracellular ligands and their plasma membrane (PM)-localized receptors is critical for the coordination of growth, development, reproduction, and responses to environmental stimuli across diverse cell types in both plants and animals. Among the various extracellular ligands, the CLE peptide family is one of the most well-characterized peptide signal families.

Among 32 CLE *Arabidopsis* family members, several CLE members have been shown to control development and stress responses in *Arabidopsis*, and cognate receptors have been identified for most of them. For instance, CLV3 is required for shoot meristem maintenance through its receptor CLV1[24]; CLE40 (also named ARABIDOPSIS CRINKLY 4, or ACR4)-CLV1 act as a ligand–receptor pair in root stem cell niche maintenance[25]; CLE45 and its receptor BAM3 play essential roles in root growth and protophloem differentiation[26]; CLE41 (also called TDIF) and PXY together regulate vascular development[27]; and the CLE45–STERILITY-REGULATING KINASE MEMBER 1 (SKM1) ligand–receptor pair functions in pollen–pistil interactions at high temperature[28]. Furthermore, CLE19 is required for root meristem maintenance[29], xylem development[29–31], embryogenesis[17], and pollen development[14]. However, its receptor has not yet been identified in any of these developmental processes; it is not even clear whether CLE19 uses the same receptors or different ones in these various tissues.

In this study, we identified the likely receptors and coreceptors of CLE19 in the regulation of pollen development. In this developmental process, CLE19 interacts directly with the extracellular LRR domain of PXL1, induces PXL1 phosphorylation, and promotes the interaction between PXL1 and its SERK coreceptors. Thus, the extracellular CLE19 signal is transduced into the intracellular signaling pathway, and subsequently, normal pollen wall formation is maintained (Figs. 1 and 5). Mutants with dysfunctional receptors or coreceptors and functionally inactive *DN-PXL1* transgenic plants all exhibited *DN-CLE19*-like pollen developmental defects, including pollen exine defects, as well as significantly increased expression of genes encoding pollen exine-related transcription factors and enzymes (Fig. 3). In addition, *DN-PXL1* strongly suppressed the *CLE19-OX*-induced pollen exine defects that occur due to missing connections in the exine network (Fig. 4). Taken together, these results strongly support the model that PXL1 and SERKs act as a receptor and coreceptors, respectively, of CLE19 to control normal pollen development and pollen exine formation.

In addition, the phylogenetic analysis (Supplementary Fig. 2) and pollen phenotypic results (Fig. 2) consistently demonstrated that *PXL1* is closely related to *PXL2* and *PXY*, and these three genes are all required or act synergistically to achieve normal pollen development. Among these three RLK proteins, both PXL1 and PXL2 directly interact with CLE19, with dissociation constants (Kd) of ~346 nM and ~714 nM, respectively (Figs. 1b and 5i), whereas PXY has only a weak interaction with CLE19 (Fig. 5j). In addition, both PXL1 and PXL2, but not PXY, could form receptor–coreceptor complexes with SERK1/2/3 proteins under the induction of CLE19 (Fig. 5 and Supplementary Figs. 7 and 8). The above results indicate that the functional mechanism mediating CLE19 signaling is well conserved between PXL1 and PXL2, but probably has diverged from PXY. One possibility is that PXY exhibits more specific binding to and is specifically activated by another CLE peptide, which has a redundant function with CLE19, in the regulation of the pollen wall. In our previous study, six anther-expressed CLE peptide played redundant roles with CLE19 in regulating pollen development, including CLE9, CLE16, CLE17, CLE41, CLE42, and CLE45[14]. Among these members, CLE41 is recognized by PXY/PXL1/PXL2 receptors to regulate vascular tissue development in the stem[15,32,33]. CLE41 could directly bind to the PXY$^{LRR}$, PXL1$^{LRR}$, and PXL2$^{LRR}$ in vitro using immunoprecipitation. ITC indicated that CLE41 binds to PXY with an affinity of 33 nM, whereas binds to PXL1 and PXL2 with affinities of 2.1 and 9.9 μM[16,32]. Together, these results revealed that PXY likely preferentially binds CLE41. In comparison, PXL1 and PXL2 are more specific for CLE19.

Thus, in addition to providing strong evidence to demonstrate that CLE19–PXL1–SERKs act as ligand–receptor–coreceptor complexes that are essential for pollen development and pollen exine formation, we provide novel evidence that the same receptor can respond to different peptide ligands in various tissues to induce distinct downstream effects in cells. The CLE peptide family is a plant-specific family with only 32 members. The relatively small number of CLE ligands compared with the much larger number (-200) of their potential receptors (such as LRR-RLKs), along with the need to regulate the functions of a variety of cells in different tissues, raises the possibility that one CLE peptide may be recognized by different RLK receptors in different tissues. Indeed, the first identified CLE member, CLV3 (CLAVATA3), interacts with three receptor complexes, CLV1, CLV2-CORYNE (CRN)/suppressor of LLP1 2 (SOL2), and RECEPTOR LIKE PROTEIN KINASE 2 (RPK2)/TOADSTOOL2 (TOAD2), to promote stem cell maintenance and shoot apical meristem differentiation, root meristem growth and differentiation, respectively[34–38]. CLE9 not only binds to HAESA-LIKE1 (HSL1), regulating cell division in the stomatal lineage, but also binds to BARELY ANY MERISTEM 1 (BAM1), regulating the periclinal cell division of xylem precursor cells[39]. Moreover, CLE40/ACR4 interacts with CLV1 and CLV2/CRN as receptors, but these interactions have the opposite effect on root meristem growth and differentiation[25]. Furthermore, CLE45 interacts with SKM1 and BAM3 and facilitates pollen tube growth and pollen–pistil interaction and protophloem differentiation in root growth[26,28,40]. CLE19 has been demonstrated to be widely required in root meristem maintenance, xylem development, embryo development, and pollen development; however, no CLE19 receptor has been identified previously in any of these developmental processes. Although PXL1 and its functionally related family members PXL2 and PXY have been demonstrated here to act as CLE19 receptors in the regulation of pollen development, the receptors that recognize CLE19 in other functional processes may be different, and this possibility remains to be investigated in the future.

## Methods

### Plant materials and growth conditions
The wild-type (WT) *Arabidopsis* used in this paper was Col-0. The *pxl1-1*, *pxl1-2*, *pxl2-1* and *pxy-3* mutants generated in Columbia background were ordered from the Arabidopsis Biological Resource Center (ABRC). The *pxl1-1 pxl2-1 pxy-3* triple mutant was kindly provided by Dr Etchells at Durham University, UK. The *serk1, serk2, bak1*, and *serk1+/− serk2−/− bak1−/−* seeds were gifts from the Weicai Yang laboratory at the Institute of Genetics and Developmental Biology, CAS, China. The *pPXL1::GUS*, *pPXL2::GUS*, and *pPXY::GUS* seeds were gifts from Jia Li at Lanzhou University, China. All mutants used in this study were genotyped by PCR analysis with the primers listed in Supplementary Table 3. All plants were cultured at 22 °C under 16 h light/8 h dark conditions.

### Generation of transgenic plants
The 1492 bp promoter region and 1–2139 bp truncated coding region of *PXL1* (*PXL1*^ΔKD^) were amplified from Col-0 genomic DNA with the primers listed in Supplementary Table 3. Each fragment was then cloned into the pDONR vector (Invitrogen) via a BP reaction followed by an LR reaction into the PGWB16 or PGWB40 vector to generate the *pPXL1::DN-PXL1-4xMYC* or *pPXL1::DN-PXL1-EYFP* constructs. Then, the constructs were transferred into the *Agrobacterium* tumefaciens GV3101 strain, and subsequently transferred to WT plants using a floral dip method to generate the *pPXL1::DN-PXL1-4xMYC* or *pPXL1::DN-PXL1-EYFP* transgenic plants.

To obtain the transgenic plants expressing the PXL1 protein, the full-length *PXL1* CDS was cloned into the pCAMBIA1306-FLAG vector (driven by the CaMV 35S promoter) using *BamHI* and *SalI*. To obtain the *pPXL1::PXL1-FLAG* transgenic plants, the 35S promoter of the pCAMBIA1306-FLAG vector was substituted by the promoter of PXL1

using *EcoRI* and *SacI*, and then the full-length *PXL1* coding region was cloned into the vector using *BamHI* and *SalI*.

For *35S::CLE19-FLAG* plants, the full length of *CLE19* coding region was cloned into pEarleyGate302 (expression driven by the CaMV 35S promoter) with the primers listed in Supplementary Table 3 to obtain the pEarleyGate302-CLE19-FLAG construct. All transgenic plants were generated using the floral dip method.

### Dot blotting assay
The PXL1-FLAG proteins were purified from 2-week-old *p35S::PXL1-FLAG* transgenic seedlings with FLAG-magnetic beads and eluted with 3xFLAG peptide (Ape-bio). PXL1-FLAG protein was blotted on a nitrocellulose membrane, and blocked with 5% nonfat milk for 1 h at 4 °C. Then, 20 µm biotin-CLE19 peptide (synthesized by Apeptide) in TBST buffer was incubated with the membrane, and anti-biotin (GNI) and anti-FLAG (GNI) antibodies were used to detect the proteins. ECL was used for chemiluminescence visualization.

### Analysis of subcellular localization by protoplasts
Full lengths coding sequences (CDS) of PXL1/PXL2/PXY were amplified via reverse transcription-polymerase chain reaction (RT-PCR) using *Arabidopsis* Columbia inflorescence cDNA as a template. Then the resulting fragments were subcloned into the pDONR/zeo entry vector and subsequently transferred into the pGWB441 binary vector. *Agrobacterium tumefaciens* strains harboring constructs expressing PXL1-YFP, PXL2-YFP, and PXY-YFP were then used to infiltrate leaves of *Nicotiana benthamiana*. After a period of 36–48 h, the infiltrated leaves were treated with enzymes solution (a mixture of 0.4% Macerozyme R-10, 1.5% Cellulase R-10, 0.4 M Mannitol, 20 mM KCl, 10 mM CaCl₂, 0.1% BSA) for 3 h to obtain protoplasts. Subsequently, these protoplasts were stained with the FM4-64 dye (10 µM in PBS) and subjected to observation by sequential excitation with 561 nm and 488 nm lasers, respectively.

### Coimmunoprecipitation assay in the tobacco transient expression system
The 1–2139 bp of PXL1 coding region (*PXL1*^ΔKD^-), 1–2094 bp of PXL2 (*PXL2*^ΔKD^), and 1–2154 bp of PXY (*PXY*^ΔKD^) were amplified with the primers listed in Supplementary Table 3 and then inserted into the pCAMBIA1306-MYC expression vector with *BamHI* and *SalI* enzyme sites to generate the pCAMBIA1306-PXL1^ΔKD^-MYC, pCAMBIA1306-PXL2^ΔKD^-MYC, and pCAMBIA1306-PXY^ΔKD^-MYC constructs, respectively. pCAMBIA1306-PXL1^ΔKD^-MYC was used as a template to obtain the pCAMBIA1306-PXL1^ΔKD-R417419A^-MYC construct by using the Mutagenesis Kit (Toyobo).

The 825 bp SERK1 (*SERK1*^ΔKD^), 834 bp SERK2 (*SERK2*^ΔKD^) and the 927 bp SERK3/BAK1 (*BAK1*^ΔKD^) coding regions were amplified with the primers listed in Supplementary Table 3. The PCR products of *SERK1*^ΔKD^, *SERK2*^ΔKD^ and *BAK1*^ΔKD^ *were* inserted into the pCAMBIA1306-GFP vector at the *KpnI* and *SalI* sites to generate the pCAMBIA1306-SERK1^ΔKD^/2/BAK1^ΔKD^-GFP constructs. pCAMBIA1306-PXL^ΔKD^-MYC, pCAMBIA1306-SERK1^ΔKD^ GFP and pCAMBIA1306-BAK1^ΔKD^-GFP were transferred into the *Agrobacterium* tumefaciens GV3101 strain.

The cultures were grown overnight in 2 x YT medium and resuspended to optical density at 600 nm (OD₆₀₀) = 2.0 in an injection buffer (150 µM acetosyringone, 10 mM MgCl₂ and 10 mM MES). The GV3101 strain with pCAMBIA1306-PXL1^ΔKD^-MYC was mixed with the strain with pCAMBIA1306-SERK1/SERK2^ΔKD^-GFP or that with pCAMBIA1306-BAK1^ΔKD^-GFP at a 1:1 ratio and infiltrated into 3-week-old *N. benthamiana* leaves. After 2 days of growth, 20 µm CLE19 was infiltrated into the expressing *N. benthamiana* leaves 3–4 h before observation, *w*ith PBS as a negative control.

Total proteins were extracted from the *N. benthamiana* leaves with the extraction buffer (50 mM Tris-HCl (pH 7.5), 150 mM NaCl, 5 mM EDTA, 5% glycerol, 1% Triton X-100 with protease inhibitor,

phosphatase inhibitor, and PMSF) and then incubated with GFP-Trap beads (Chromo Tek) at 4 °C for 4 h, followed by two washes (50 mM Tris-HCl (pH 7.5), 150 mM NaCl, 5 mM EDTA, 5% glycerol, and 0.01% Triton X-100). Proteins were separated on SDS-PAGE gels and detected by anti-Myc (GNI) antibody via western blots.

## Quantitative real-time PCR

We used qRT–PCR to analyze the expression levels of *PXL1*, *PXL2* and *PXY* in the corresponding T-DNA mutants, the PXL1$^{LRR}$ level in *DN-PXL1* transgenic plants, the tapetum development- and pollen exine formation-related gene expression levels in *DN-PXL1* and *CLE19-OX/ DN-PXL1* transgenic plants, and the CLE19 expression level in transgenic mutants. Total RNA was extracted from the inflorescences of the corresponding plants using an RNAiso Plus Kit (Takara). Then, 1 μg of total RNA was reverse transcribed using PrimeScript II Reverse Transcriptase (Takara). qRT–PCR was performed using SYBR premix Ex Taq II (Takara) and a Bio-Rad CFX96 Real-Time PCR detection system. Primer information is shown in Supplementary Table 3.

## LCI assay

The 2139 bp PXL1 coding region (*PXL1$^{\Delta KD}$*) was amplified with the primers listed in Table S3 and then inserted into the expression vector pNL with NLuc at the C-terminus by using a recombination kit (Vazyme). The 825 bp *SERK1* (*SERK1$^{\Delta KD}$*) and the 927 bp *BAK1* (*BAK1$^{\Delta KD}$*) coding regions were amplified with the primers listed in Supplementary Table 3. The PCR products of *SERK1$^{\Delta KD}$* and *BAK1$^{\Delta KD}$* were inserted into the expression vector pCL with CLuc at the C-terminus by recombination kit (Vazyme). The plasmids were introduced into Agrobacterium strain GV3101, and then single colonies were cultured in liquid YT medium. The OD values were measured with a UV/Vis spectrophotometer (Eppendorf). Then, resuspensions of well-cultured Agrobacterium containing the intended plasmid were mixed with 1:1 ratio as required and the different combinations were infiltrated into 4-week-old *N. benthamiana* leaves. After 2 days, 20 μm CLE19 was infiltrated into the expressing *N. benthamiana* leaves, with PBS as a negative control, 3–4 h before observation. Then, each infiltrated leaf was sprayed with 0.25 mM luciferin substrate and placed in the dark for 10 min, and LB985 NightShade (Berthhold Tech) with IndiGo software was used to detect the interaction signal.

## In vivo phosphorylation assay

Ten-day-old *pPXL1:PXL1-FLAG* transgenic seedlings were tested to observe the phosphorylation status of PXL1. The seedlings were cultured in 1/2 MS liquid medium treated with mock or 20 μM CLE19, CLE19$_{G6T}$ or CLV3 peptide for 1.5 h and then ground to a fine powder in liquid N$_2$ for protein extraction performed as described above. The extracted protein was analyzed by Phos-tag (Wako) gel and immunoblotting using anti-FLAG (GNI) antibody. The anti-HSP70 (Abmart) antibody was used to determine the amount of input protein. Anti-FLAG beads (Abmart) were used for immunoprecipitation to purify the PXL1-FLAG protein. Anti-pS/T (ECM) was used to detect the phosphorylation status of the PXL1 protein after CLE19 treatment. The seedling protein extracted was also treated with λpp (NEB) at 30 °C for 0.5 h and then analyzed by anti-FLAG and anti-HSP antibodies.

## Protein expression and purification

The coding regions corresponding to the PXL1 LRR domains (residues 1–639) or SERK1 LRR domains (residues 1–213) were subcloned into the pFastBac1 (Invitrogen) vector to generate the PXL1$^{LRR}$–6×His or SERK1$^{LRR}$–6×His constructs. Both fusion proteins were expressed in High Five cells at 22 °C and purified using an Ni-NTA (Novagen) column. After elution and concentration, the proteins were further purified by size-exclusion chromatography (HiLoad 200, GE Healthcare) in buffer containing 10 mM Bis-Tris, pH 6.0, and 100 mM NaCl.

## Gel filtration–mass spectrometry and gel filtration assays

Purified PXL1$^{LRR}$ protein (with buffer containing 10 mM Bis-Tris pH 6.0, 100 mM NaCl) was incubated with synthesized peptide mixture (0.1 mg for each; Supplementary Table 1) on ice for 1 h. Then, each mixture was subjected to gel filtration (Superdex 200, GE Healthcare) for analysis. The relevant peak fractions of PXL1$^{LRR}$ were collected and 2 μl was used for analyzed by MALDI-TOF–mass spectrometry analysis.

The purified PXL1$^{LRR}$ and SERK1$^{LRR}$ proteins were incubated with or without CLE19 peptide on ice for 1 h. Then, the mixtures were separated by gel filtration (Hiload 200, GE Healthcare) and the peak fractions were used for SDS-PAGE followed by Coomassie blue staining.

## ITC assay

ITC200 was used to quantify the binding affinity of CLE19 with PXL1$^{LRR}$ protein. The sample was prepared in a buffer containing 10 mM Bis-Tris, pH 6.0, and 100 mM NaCl and titration was executed at 25 °C. 0.1 mM PXL1$^{LRR}$ was titrated against 1 mM CLE19. The ITC data was analyzed using MicroCal Origin 7.0.

## Accession numbers

The sequences of genes mentioned in this paper can be download from the GenBank/EMBL database under the following accession number: *PXL1* (AT1G08590), *PXL2* (AT4G28650), *PXY* (AT5G61480), *CLE19* (AT3G24225), *SERK1* (AT1G71830), *BAK1* (AT4G33430). Germplasm used included: *pxl1-1* (SALK_001782), *pxl1-2* (SALK_128519), *pxl2-1* (SALK_114354), *pxy-3 (SALK_026128)*, *serk1* (SALK_071511), *serk2* (SALK_058020), and *bak1* (SALK_116202).

## Reporting summary

Further information on research design is available in the Nature Portfolio Reporting Summary linked to this article.

# Data availability

The data supporting the findings in this study are available and described within the paper and its Supplementary Information. Source data are provided with this paper.

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

## Acknowledgements

We thank Dr. Jia Li at Lanzhou University for *pPXL1::GUS, pPXL2::GUS,* and *pPXY::GUS* transgenic seeds; Dr. Weicai Yang at Institute of Genetics and Developmental Biology in CAS for *serk1+/− serk2 bak1* mutant seeds; and Dr. J. Peter Etchells in Durham University for *pxl1-1 pxl2-1 pxy-3* triple mutant seeds. This work was supported by the grant from the Ministry of Science and Technology, People's Republic of China (2021YFA0909303), grants from the National Natural Science Foundation of China (31822005, 31870294 and 31670316), and grant from the 2115 Talent Development Program of China Agricultural University (to W.S.).

## Author contributions

F.C., J.C., H.M. and W.S. conceived the projects. F.C., Y.Y. and W.S. designed the experiments and analyzed the data. Y.Y., W.S., N.Z., S.Z., W.L. and S.W. performed the experiments. J.W. carried out the phylogeny analysis under the supervision of C.H. F.C., Y.Y. and W.S. draft the manuscript. F.C. and H.M. revised and finalized the manuscript. All authors read and approved the manuscript.

## Competing interests

The authors declare no competing interests.
