## [Peer Review File · Nature Communications]

PXL1 and SERKs act as receptor–coreceptor complexes for the CLE19 peptide to regulate pollen developmentReviewer #1 (Remarks to the Author):

The authors report the identification of PXL1 and SERKs, as putative receptors and co-receptors, respectively, for the CLE19 peptide, and involved in the regulation of tapetum transcriptional activity, essential for proper pollen development.

There are some issues that need to be addressed to make suitable for publication. Here are my comments:

I would reorganize the entire paper. It makes no sense to analyze GUS expression, count pollen grains and show exine malformations first for the pxl1 mutants and then repeat the same for pxl2, pxlY and the triple mutant.

Line 110. What other LRR domains besides PXL1 were used in this assay? Are they mentioned in the text? Was the CLE19-PXL1 complex the only one identified? Was CLE19 the only CLE peptide that co-purified with PXL1? This is important considering that in a previous article of the group it was shown that in addition to CLE9, CLE16, CLE17, CLE19, CLE41, CLE42 and CLE45 are involved in another development.

Figure 1D. Specify what do the quantification numbers shown below each of the gels correspond to.

Line 136. Remove the word "specifically" as it cannot be ensured that the CLE19-PXL1 interaction is specific due to the fact that no other RKs were tested.

Figure 2A: How was the location study carried out? Is this a transient expression in Benthamiana leaves? Please specify. To ensure that PXL1 is effectively expressed in PM, a plasmolysis assay using FM4-64 staining should be performed.

Line 143. The GUS assay indicates tissue expression, not "biological function." Please modify this sentence.

Figures 2C-D and S1C. Please provide better pictures of the inflorescence and the anthers.

Line 166 and others. Remove all "obviously" terms from the text.

Line 167. Delete the expression ">350" and give the exact number of pollen grains counted for each genotype.

Figure S2: Which is the "n" of the results shown in this figure?

Figure 3B. Please provide a statistical analysis for this figure.

Figure 3. Explain why there is no correlation between the expression level of the transgenes and the severity of the pollen exine defects.

Lines 303-306. Move this paragraph to the discussion section.

Line 320. The figure is S9A not S8A. Fix it.

Line 321. The authors show that the serk1+/- serk2-/-bak1-/- mutant has only 122 pollen grains. How does it compare to the triple mutant pxl? This is important because the authors suggest that SERKs would serve as coreceptors; however, it appears that the effect is stronger for SERKs than for PXLs.

Figures 5E-F. Include molecular weight reference markers for gel filtration assay. Explain why the PXL1 elution fractions on the gel filtration column do not change in the presence of CLE19+SERK1. PXL1 always elutes in fractions 66-72. If a protein complex is present, the elution of PXL1 should shift to earlier fractions corresponding to higher molecular weights.

Reviewer #2 (Remarks to the Author):

This manuscript by Yu, Song and Zhai examines the molecular control of exine formation during pollen development. Starting with the known role of the peptide CLE19 in exine regulation, they isolate PXL1 as a potential receptor, demonstrating its phosphorylation in the presence of CLE19. The pollen developmental phenotypes of dominant negative PXL1 mutants and pxl1 T-DNA insertion mutants is shown to be similar to that of CLE19 dominant negative plants. The authors go on to show similar phenotypes in pxy1 and pxl2 mutants, indicating that these kinases may act as alternative receptors for CLE19. It is unclear why the authors then continue to focus on PXL1, rather than incorporating PXY and PXL2 further into the experiments. In the last part of the study, the authors make an educated guess that SERKs/BAK1 may act as co-receptors of PXL1, further demonstrating pollen phenotypes in serks/bak1 and an interaction mediated by CLE19 between PXL1 and SERK1 or BAK2.

Overall, it is a nice study that is generally well performed, and should be of general interest to plant biologists studying molecular signalling mechanism. The data looks sound and the study well designed, however there are some omissions of data, statistics and descriptions which should be moderately easy to fix.

I therefore have a number of comments which can mostly be addressed through adjustment of the text and figures:

1. Are PXY and PXL2 also able to form complexes with SERK1 and BAK1, and are they able to bind CLE19? Does SERK2 also function as a co-receptor with PXY/PXL1/PXL2? Additionally, can PXL1 form homodimers or heterodimers with PXY and PXL2? Although I don't think these experiments are absolutely required for publication, they would make the study more complete. Alternatively, greater speculation on how this complex might function would be desirable in the discussion.
2. For many of the experiments, the number of replicates performed is not clear. The number of needs to be explicitly stated in the manuscript, including in figure legends.
3. Some of the figures are missing the statistical tests needed to definitively conclude whether the phenotypes seen are meaningful. e.g. chi-squared tests or another appropriate statistical test for Figs 3E and 4D, S7D. For these sort of figures, it would also be normal to include how many pollen grains were phenotyped.
4. Line 86 – 'CLE19 and other CLEs act as brakes to limit AMS expression' – which other CLEs? Is it possible these CLEs are also bound by PXY/PXL1/PXL2? This could be added to the discussion.
5. Line 120 – for readers less familiar with the field, it might be helpful to directly state why interactions were tested between CLE19 and only the LRR domains of PXL1
6. I would be hesitant about calling a peptide a hormone. A peptide signal might be a better way of phrasing this.
7. Fig 1 panel A left diagram needs further explanation. What are the coloured shapes? Is this a schematic of the experiment, or actual results? I presume this is a schematic, however it needs further explanation in the figure legend.
8. Fig 1 Panel B – it might be preferable if the legend refers to 'PXL1 and CLE19' rather than 'PXL1-CLE19' since a dash is otherwise used to indicate a fusion protein. Also, should specify plant species in the figure legend.
9. Need to explain in text why CLE19G6T is assumed or known to be functionally inactive.
10. Fig 1E and D – should label with size of proteins/ladder bands as an indicator of size.
11. Fig 1D - Need to state what the 0.57 is/how it has been calculated.

12. Fig 1E - Gel bands trending up for anti-FLAG and down for anti-HSP – this seems unlikely if both panels have been taken from the same gel. Could the authors please share an image of the entire blot with the reviewers?
13. Fig 2 A/B – here it would be helpful to also show the brightfield image without the YFP overlay. It is very hard to see the cell boundaries in the merged image.
14. Fig S1C – I don't find this image convincing – it is very hard to make out GUS staining on the image.
15. Fig 2F – while the qPCR shows low expression of PXL1 in the pxl1 T-DNA insertion lines, these primers span the T-DNA insertion site. To show that there are low levels of transcript throughout and therefore that a truncated protein is unlikely to be produced, primers that are 5' of the insertion site should also be used for qPCR.
16. Fig 3B, F and G (and others) SD or SEM on qPCRs seem surprisingly small for biological samples. Are these calculated on technical replicates rather than biological replicates? Also need to specify whether SEM or SD is shown for.
17. Fig 3E – it is incorrect to describe this as 'statistical results'. Proportion of pollen grains by phenotype might be a better way to express this.
18. Fig S6 – please add in qPCR primer locations.
19. Lines 234 to 238 'To test this idea, pxl2-1, pxy-3, and pxy-5 single knockout mutants (Fig. S6) were analyzed for their pollen developmental phenotypes. Anthers from all three single mutants were slightly smaller, and their pollen counts in each anther was obviously reduced as well, similar to those of pxl1-1 and pxl1-2.' This data is not shown in Fig S6, nor could I find it elsewhere in the paper – this should be added to support the statement.
20. Fig 4 – figure legend describes panels A and B in opposite order to what is shown.
21. Arrows, asterisks, boxes are often shown on figures with no additional explanation of what they mark given in the figure legend. In Fig S7A, arrow heads are VERY small and hard to see.
22. Line 250 – What is known of MOL1's function and why is it of particular note in the phylogenetic tree in Fig S8? This should be added to the text.
23. Line 255 – amino acid similarity between PXL1 and PXL2 is commented on. What of PXY compared to PXL1 and PXL2?
24. Line 261 – It would be helpful to remind the reader what the developmental defects of the CLE19-OE line are.
25. Line 302 – I don't understand what is meant here '...., indicating that the CLE19 signal and CLE19 signal through PXL1 are (at least partially) dependent on the presence of a functional PXL1 receptor.' It might be advisable to shorten the sentence and make the meaning clearer.
26. Line 320 – 'serk1+/- serk2-/-bak1-/-' Please be consistent with how you refer to this line in the manuscript.
27. Line 320 – I believe Fig S9A is the correct figure to be cited here. Also, while serk1 is referred to in the text as having been pollen phenotyped, this is not included in the figure. For completeness, it should be included.
28. Fig 5 – order of panels could be improved.
29. Line 401 – I think it would be more correct to say that a peptide is recognised by the receptor

than vice versa, or to say that the peptide can bind to different receptors.

30. What about pollen phenotypes of PXY/PXL double mutants? Are these intermediate to single and triple phenotypes?

31. Light levels and type of illumination should be included in the description of the plant growth conditions, along with type of soil used.

32. pGWB vectors – don't these usually have three digit codes where the first number indicated the selection marker? Please include the full number.

33. In figure legends, please be clear on which experiments were performed in Arabidopsis and which in tobacco.

34. Please specify Agrobacterium strain used for all plasmids transformed into Arabidopsis or tobacco.

35. Line 486-489 'To analyze the expression level of PXL1, PXL2 and PXY in the corresponding T-DNA mutants, PXL1LRR level in DN-PXL1 transgenic plant, tapetum development and pollen exine formation related genes expression level in DN-PXL1 and CLE19-OX/DN-PXL1 transgenic plants, CLE19 expression level in transgenic mutants.' I am unclear on your meaning here. Please rephrase.

36. There are a few grammatical and spelling errors that could do with correction. This mostly affects the materials and methods and figure legends, however there are also a few errors elsewhere in the manuscript.

Reviewer #3 (Remarks to the Author):

Peptide hormones are important factors involved in diverse developmental and physiological processes. In this study, the authors searched the cognate receptors that recognize CLE19 peptide by a library-based interaction ability screening. The primary findings of this study are the identification of the PXL1 and related receptors as well as co-receptors for CLE19 peptide, and the module plays a role in the regulation of pollen development. The research plans, especially genetic analyses seem well organized and the results will be of interest to readers interested in plant physiology and small molecule signaling peptides. I very much enjoyed reading this manuscript and I think that I can recommend this for publication from Nature communications. However, the current version manuscript contains several concerns and needs to be improved before acceptance.

Most importantly, many of the WB photos are of poor quality hence the results are not convincing. Particularly, in Fig. 1D, the important phosphorylated PXL1 signal is not very clear, similarly, in Fig 1E, the loading control varies greatly. The authors should consider replacing the pictures with ones that better depict the results.

Next, the authors claim that PXL1 is the cognate receptor for CLE19, which has already been reported as the receptor for TDIF. Genetic data shown in this manuscript adequately support this hypothesis, but there is insufficient discussion of the similarities and differences between CLE19 and TDIF.

In this context, showing sequence alignments and comparison of the peptide hormones, namely CLE19 and TDIF will be valuable. In addition, since the crystal structure of TDIF-PXY has already been reported, it is desirable to make a comparison with it. Specifically, comments should be made regarding the amino acid residues responsible for the interaction.

Introducing mutation(s) to the interaction domain of PXY for the TDIF binding residues should also be tested for the ability to interact with CLE19. Is the position of the DN mutation in CLE19

explainable to the opposing effects?

For SERKs, the main argument of the authors that SERKS are acting as co-receptor for CLE19-PXL1 is based on the mutant phenotypes as well as biochemical experiments. Although the genetics provided in this manuscript is very sophisticated, the epistasis between SERKs and CLE19 or PXY is not fully proved. Although I think the regulatory mechanisms that involve the CLE19-PXY-SERK module is a promising hypothesis, evidence by genetic analysis of SERKs and CLE19 or PXY is desirable to establish a convincing model.

For RT-PCR analyses of pollen exine formation genes, as the context of tissue-specificity is important to this study, analysis by bulk RNA alone is not sufficient. It is desirable to observe the markers for (selected) genes.

It will be interesting to discuss whether CLE19 binding and TDIF binding are similar or different stimuli for receptors. This is entirely optional, function replacement experiments with CLE19p:TDIF and/or reciprocal replacement may help this discussion.

minor point

L338 "interaction signal"

I think the word is inappropriate (overstatement or misleading). The signal itself indicates the MYC signal or PXL1-MYC, and the presence of the signal suggests the interaction.

typos

L338 CLE19treatment

need space

Point-by-Point Responses to Referees

Reviewer #1 (Remarks to the Author):

The authors report the identification of PXL1 and SERKs, as putative receptors and co-receptors, respectively, for the CLE19 peptide, and involved in the regulation of tapetum transcriptional activity, essential for proper pollen development.

There are some issues that need to be addressed to make suitable for publication. Here are my comments:

Response: We greatly appreciate the encouraging comments and constructive suggestions from the referee. In the revised manuscript, we addressed all these concerns.

I would reorganize the entire paper. It makes no sense to analyze GUS expression, count pollen grains and show exine malformations first for the *pxl1* mutants and then repeat the same for *pxl2*, *pxy* and the triple mutant.

Response: Thank the referee for this point. As suggested, we have reorganized the entire manuscript (Line 144-200).

We now first described the phenotypic observations of *pxl1* single mutants immediately after the description of the CLE19-PXL1 interaction. Then, we show the sequence similarity and evolutionary relationship, GUS expression in anthers, and PM localization (GFP distribution) of these three RLK proteins. Finally, we presented the phenotypic analyses of the single, double and triple mutants of PXL1/PXL2/PXY.

Fig. 2 and Fig. S1-5 have been rearranged accordingly.

Response Fig. 1. (the same as the revised Fig. 2). Phenotypic analyses, showing PXL1, PXL2 and PXY together are required for pollen development. (a-c) Phenotypic observations of anther and pollen of WT, *pxl1*, *pxl2*, *pxy* single and *pxl1 pxl2*, *pxl2 pxy* double and *pxl1 pxl2 pxy* triple mutants. (a) Alexander stained anthers and (b-c) SEM images of pollen and pollen exines are shown. Bar =100 μ m for (a), 10 μ m for (b), and 2 μ m for (c), The red stars indicate the pollen exine defect. (d) Statistical results for the pollen amounts

per anther of various genotypes, p values were calculated by T-Tests, the n values showed the anthers' number counted. (e) SEM images showing the two types of defective pollen exines, which we defined as moderate-D and severe-D. Bar = 5 μm for the upper pictures, bar = 4 μm for the bottom enlarged pictures. (f) Statistical results show the portion of pollen with normal, moderate-D and severe-D exine defects from various genotypes. P values are calculated by chi-square, and n indicates anther count used for each genotype. (g) Expression of *PXL1*, *PXL2* and *PXY* genes in inflorescences, anthers and anther lobes using *PXL1::GUS*, *PXL2::GUS* and *PXY::GUS* reporter lines. bar =100 μm for inflorescences, 50 μm for anthers, 10 μm for anther lobes. The expression in early floral buds is indicated with red arrowhead, and the anther tapetum layer in each lobe is indicated by dotted lines. (h) Subcellular localization of *PXL1-YFP*, *PXL2-YFP* and *PXY-YFP* were observed in a transient expression system of *Nicotiana benthamiana* leaf cells and protoplasts. YFP signal is shown in green, and FM4-64 staining is shown in red. bar = 5 μm .

Line 110. What other LRR domains besides PXL1 were used in this assay? Are they mentioned in the text? Was the CLE19-PXL1 complex the only one identified? Was CLE19 the only CLE peptide that co-purified with PXL1? This is important considering that in a previous article of the group it was shown that in addition to CLE9, CLE16, CLE17, CLE19, CLE41, CLE42 and CLE45 are involved in the anther development.

Response: Thank the referee for this excellent comment. We previously reported that the LRR domains of *RGFR1*, *HSL2* and *PXY* were able to bind to the *RGF1*, *IDA*, and *CLE41* peptides respectively, in the gel-filtration MS assay¹⁻³. Therefore, the *CLE19-PXL1* complex was not the only peptide–receptor pair identified using this approach.

However, in the assay of current study with *CLV3*, *CLE3*, *CLE6*, *CLE8*, *CLE9*, *CLE11*, *CLE12*, *CLE14*, *CLE16*, *CLE18*, *CLE19*, *CLE20*, *CLE25*, *CLE40*, *CLE41/TDIF*, *CLE43*, *CLE45*, and *CLE46* (Supplemental Dataset 1), *CLE19* was the only CLE peptide that co-purified with *PXL1*, indicating that *CLE19* binds *PXL1* with much higher binding affinity compared to other CLE peptides.

Consistent with this observation, *CLE41/TDIF* was reported to bind to *PXL1* with an affinity of 2 μM (Fig. S4 of the ref. 3)³, which is 6-fold lower than the affinity of *CLE19* for *PXL1* (0.35 μM) (Fig. 1b). We have included the following information in Lines 369-384 of the revised discussion.

“Among these three RLK proteins, both *PXL1* and *PXL2* directly interact with *CLE19*, with dissociation constants (*K_d*) of ~346 nM and ~714 nM, respectively (Fig. 1b and 5i), whereas *PXY* has only a weak interaction with *CLE19* (Fig. 5j). In addition, both *PXL1* and *PXL2*, but not *PXY*, could form receptor–coreceptor complexes with *SERK1/2/3* proteins with the induction of *CLE19* (Fig. 5 and Fig. S7-8). The above results indicate that the functional mechanism mediating *CLE19* signalling is well conserved between *PXL1* and *PXL2*, but probably has diverged from *PXY*. One possibility is that *PXY* exhibits more specific binding to and is specifically activated by another CLE peptide, which has a redundant function with *CLE19*, in the regulation of the pollen wall. In our previous study, six anther-expressed CLE peptides played redundant roles with *CLE19* in regulating pollen development, including *CLE9*, *CLE16*, *CLE17*, *CLE41*, *CLE42*, and *CLE45*¹². Among these members, *CLE41* is recognized by *PXY/PXL1/PXL2* receptors to regulate vascular tissue development in the stem^{13,31,32}. *CLE41* could directly bind to the *PXY*^{LRR}, *PXL1*^{LRR}, and *PXL2*^{LRR} in vitro using immunoprecipitation. ITC indicated that *CLE41* binds to *PXY* with an affinity of 33 nM, but it binds to *PXL1* and *PXL2* with affinities of 2.1 μM and 9.9 μM ^{14,31}. Together, these results revealed that *PXY* likely preferentially binds *CLE41*. In comparison, *PXL1* and *PXL2* are more specific for *CLE19*.”

Supplemental Dataset 1. CLE peptide used in this study	
Peptide	Sequence (from N terminus to C terminus)
CLV3	RTV(Hyp)SG(Hyp)DPLHH
CLE3	RLSPGG(Hyp)DPRHH
CLE6	RVSPGG(Hyp)DPQHH
CLE8	RRVPTG(Hyp)NPLHH
CLE9	RLV(Hyp)SG(Hyp)NPLHN
CLE11	RVVPSG(Hyp)NPLHH
CLE12	RRVPSG(Hyp)NPLHH
CLE14	RLVPKG(Hyp)NPLHN
CLE16	RLVHTG(Hyp)NPLHN
CLE18	RQIPTG(Hyp)DPLHN
CLE19	RVIPTG(Hyp)NPLHN
CLE19 _{G6T}	RVIPTNPLHN
CLE20	RKVKTGSNPLHN
CLE25	RKVPNG(Hyp)DPIHN
CLE40	RQVPTGSDPLHH
CLE41/TDIF	HEV(Hyp)SG(Hyp)NPISN
CLE43	RRIPSS(Hyp)DRLHN
CLE45	RRVRRGSDPIHN
CLE46	HKHPSG(Hyp)NPTGN

Reference:

1. Song, W., Liu, L., Wang, J. et al. Signature motif-guided identification of receptors for peptide hormones essential for root meristem growth. *Cell Res.* **26**, 674–685 (2016).
2. Qian, P., Song, W., Yokoo, T. et al. The CLE9/10 secretory peptide regulates stomatal and vascular development through distinct receptors. *Nature Plants.* **4**, 1071–1081 (2018).
3. Zhang, H., Lin, X., Han, Z. et al. Crystal structure of PXY-TDIF complex reveals a conserved recognition mechanism among CLE peptide-receptor pairs. *Cell Res.* **26**, 543–555 (2016).

Figure 1D. Specify what do the quantification numbers shown below each of the gels correspond to.

Response: Thanks for the referee's comment. These quantification numbers indicate the amount of proteins in each lane. The above description has been added into the figure legend.

Line 136. Remove the word "specifically" as it cannot be ensured that the CLE19-PXL1 interaction is specific due to the fact that no other RKs were tested.

Response: Thanks for the advice. The word "specifically" was deleted accordingly.

Figure 2A: How was the location study carried out? Is this a transient expression in Benthamiana leaves? Please specify. To ensure that PXL1 is effectively expressed in PM, assay using FM4-64 staining should be performed.

Response: Thanks for the referee's comment. The PXL1 localization analysis presented in Figure 2A of the previous version (Figure 2g and Figure S3 of the revised manuscript) was carried out using the *Nicotiana benthamiana* leaf transient expression system. To ensure that the PXL1 was effectively expressed in the PM, we obtained protoplasts with PXL1-GFP expression and labelled the protoplast PM with FM4-64, as suggested by the referee. The results revealed that PXL1-GFP and FM4-64 are colocalized with each other (Fig. 2h of the revised manuscript). The same analyses were also carried out for PXL2 and PXY.

Response Fig. 2. Subcellular localization of PXL1-YFP, PXL2-YFP and PXY-YFP were observed in a transient expression system of *Nicotiana benthamiana* leaf cells and protoplasts. YFP signal is shown in green, and FM4-64 staining is shown in red. bar = 5 μ m. (shown as Fig. 2h in the revised manuscript).

Line 143. The GUS assay indicates tissue expression, not "biological function." Please modify this sentence.

Response: Thanks for pointing out this. We have modified accordingly in the text.

Figures 2C-D and S1C. Please provide better pictures of the inflorescence and the anthers.

Response: Thanks for the referee's comment. In the revision, we have included photographs of the inflorescence and the anthers with better quality. (Please see the revised Figure 2g).

Line 166 and others. Remove all "obviously" terms from the text.

Response: We have removed the word "obviously" from the text.

Line 167. Delete the expression ">350" and give the exact number of pollen grains counted for each genotype.

Response: We have provided the exact number of pollen grains for each genotype in the revised manuscript (Line 152-154) as following:

"In comparison to those of the wild type (WT), the anthers of *pxl1-1* and *pxl1-2* were slightly smaller, and the pollen count in each anther was reduced from 461 in WT anthers down to 348 in *pxl1-1* and 355 in *pxl1-2* (Fig. 2a and d)."

Figure S2: Which is the "n" of the results shown in this figure?

Response: The "n" indicates the number of anthers counted for each genotype. We have added this information "n indicates the number of anthers counted" to the figure legends of the Fig. 2d and Fig. 3f of the revised manuscript.

Figure 3B. Please provide a statistical analysis for this figure.

Response: As suggested, we have performed the statistical analysis for this figure (current Figure S5b-d).

Figure 3. Explain why there is no correlation between the expression level of the transgenes and the severity of the pollen exine defects.

Responses: Thank the referee for this comment. The situation is somewhat complex, as explained below.

We analyzed four transgenic lines carrying *DN-PXL1* driven by the native promoter, two with the MYC tag (*DM#8*, *DM#25*) and two with the YFP tag (*DE#12*, *DE#20*). For the two MYC tag transgenic lines, the expression level of *DN-PXL1* in *DM#25* was higher than that in *DM#8*. Consistently, the pollen exine defects of *DM#25* were statistically more severe than those of *DM#8*. Similarly, for the two EYFP tag transgenic lines (*DE#12* and *DE#20*), the expression level of *DN-PXL1* was also correlated with the severity of the pollen exine defects (Fig. 3g and Fig. S5, also shown in the following); even though the number of pollen grains was slightly more in *DE#12* than *DE#20*, the difference was not significant. Thus, the severity of the pollen exine defects is correlated with the expression level of *DN-PXL1*, when the two MYC-tagged transgenic lines are considered separately from the two YFP-tagged transgenic lines. It is possible that different fusion proteins have some differences in protein activities.

We have added the above explanation in the manuscript on Lines 204-210.

Response Fig.3. (a) Relative expression level of DN-PXL1 in the WT, *DM#8*, *DM#25*, *DE#12* and *DE#20*. (b) Portion of pollen with normal, moderate-D and severe-D exine defects of the WT, *DM#8*, *DM#25*, *DE#12* and *DE#20*. P-value is calculated by chi-square. (a) was the same as Fig. S5d, and (b) was the same as Fig. 3g in the revised manuscript.

Lines 303-306. Move this paragraph to the discussion section.

Response: Thanks. This paragraph has been moved to the discussion section in the revised manuscript.

Line 320. The figure is S9A not S8A. Fix it.

Response: Thanks. The error has been fixed.

Line 321. The authors show that the *serk1+/- serk2-/-bak1-/-* mutant has only 122 pollen grains. How does it compare to the triple mutant *pxl1*? This is important because the authors suggest that SERKs would serve as coreceptors; however, it appears that the effect is stronger for SERKs than for PXLs.

Response: We thank the referee for this important comment. We agree that the reduction in pollen count of the *serk1+/- serk2-/-bak1-/-* mutant is greater than that of the *pxl1 pxl2 pxy* triple mutant. In comparison to the WT, which produces ~460 pollen grains per anther, the *serk1+/- serk2-/-bak1-/-* mutant produces only ~122 (Fig. S6b) and the *pxl1 pxl2 pxy* triple mutant has ~323 in each anther (Fig. 2d).

However, the proportion of pollen grains with abnormal pollen exine was similar in the *serk1+/- serk2-/- bak1-/-* and the *pxl1 pxl2 pxy* triple mutants. Only 3% of pollen grains show abnormal pollen exine structure in WT anthers. In comparison, the proportion was increased to 25.7% for the *serk1+/- serk2-/-bak1-/-* mutant (Fig. S6c) and to 21% for the *pxl1 pxl2 pxy* triple mutant (Fig. 2f). The similarity of pollen exine defects between *serk1+/- serk2-/-bak1-/-* and *pxl1 pxl2 pxy* mutants, together with the biochemical evidence that PXL1 and PXL2 directly interact with SERK1/2/3 under the induction of CLE19, strongly supports the idea that SERKs and PXLs act as coreceptor and receptor complexes of CLE19.

Because the *serk1+/- serk2-/-bak1-/-* mutant produces much less pollen than the *pxl1 pxl2 pxy* mutant does, our explanation is that SERK1/SERK2 also play important roles in regulating the development of the tapetum and microspores. The SERK1/SERK2 proteins have been reported to act as coreceptors of EMS1 to perceive the TPD peptide signalling in regulating the cell fate determination of the tapetum layer and the maturation of microspores¹⁻³. The *serk1 serk2* double homozygous mutant is male sterile and lack the tapetum. Although we used a heterozygous of *serk1* in the triple mutant, we still believe that the dysfunction of half of the *SERK1* proteins, together with *SERK2*, partially affected the development of the tapetum and microspores and finally reduced the yield of pollen grains.

We have added a brief discussion related to these results on Lines 281-283: “with only approximately 122 pollen grains in each anther (Fig. S6b), **probably because of that the dysfunction of half of the *SERK1* proteins, together with *SERK2*, partially affected the development of the tapetum and microspores and finally reduced the yield of pollen grains. Therefore, we focused our phenotypic observation on the pollen exine.**”

References:

1. Albrecht, C., Russinova, E., Hecht, V., Baaijens, E. & De, V. S. The *Arabidopsis thaliana* SOMATIC EMBRYOGENESIS RECEPTOR-LIKE KINASES1 and 2 control male sporogenesis. *The Plant Cell*. **17**, 3337-3349 (2006).
2. Colcombet J., Boisson-Dernier A., Ros-Palau R., Vera C.E., and Schroeder J.I. *Arabidopsis* SOMATIC EMBRYOGENESIS RECEPTOR KINASES1 and 2 Are Essential for Tapetum Development and Microspore Maturation. *The Plant Cell*. 3350-3361 (2005).
3. Li Z., Wang Y., Huang J. Ahsan. N., et al., Two SERK Receptor-Like Kinases Interact with EMS1 to Control Anther Cell Fate Determination. *Plant Physiol.* **173**, 326-337 (2017).

Figures 5E-F. Include molecular weight reference markers for gel filtration assay. Explain why the PXL1 elution fractions on the gel filtration column do not change in the presence of CLE19+SERK1. PXL1 always elutes in fractions 66-72. If a protein complex is present, the elution of PXL1 should shift to earlier fractions corresponding to higher molecular weights.

Response: Thanks for the comment. The molecular weight reference makers have been included. The peak elution volume of PXL1-CLE19-SERK1 shifted forward by one fraction compared to that of PXL1, which was consistent with the previous report that 1-2 fractions were shifted forward of PXY-TDIF-SERK1 compared to PXY alone¹. One possible reason is that the SERK protein binding to the lateral side of PXY/PXL1 LRR domain does not significantly alter the maximum diameter of the protein molecule, which is the major determinant of elution volume in gel filtration assays.

Although the PXL1-CLE19-SERK1 complex shifted forward by only one fraction, the SERK protein (which is smaller than the PXL1 protein) shifted forward by 7 fractions, which strongly indicated that SERK1 forms a protein complex with PXL1 in the presence of CLE19.

References:

1. Zhang, H., Lin, X., Han, Z. et al. Crystal structure of PXY-TDIF complex reveals a conserved recognition mechanism among CLE peptide-receptor pairs. *Cell Res.* **26**, 543–555 (2016).

Again, we appreciate all comments and suggestions from this referee that help us to improve our manuscript, which is more readable to the audience.

Reviewer #2 (Remarks to the Author):

This manuscript by Yu, Song and Zhai examines the molecular control of exine formation during pollen development. Starting with the known role of the peptide CLE19 in exine regulation, they isolate PXL1 as a potential receptor, demonstrating its phosphorylation in the presence of CLE19. The pollen developmental phenotypes of dominant negative PXL1 mutants and pxl1 T-DNA insertion mutants is shown to be similar to that of CLE19 dominant negative plants. The authors go on to show similar phenotypes in pxy1 and pxl2 mutants, indicating that these kinases may act as alternative receptors for CLE19. It is unclear why the authors then continue to focus on PXL1, rather than incorporating PXY and PXL2 further into the experiments. In the last part of the study, the authors make an educated guess that SERKs/BAK1 may act as co-receptors of PXL1, further demonstrating pollen phenotypes in serks/bak1 and an interaction mediated by CLE19 between PXL1 and SERK1 or BAK2.

Overall, it is a nice study that is generally well performed, and should be of general interest to plant biologists studying molecular signalling mechanism. The data looks sound and the study well designed, however there are some omissions of data, statistics and descriptions which should be moderately easy to fix.

I therefore have a number of comments which can mostly be addressed through adjustment of the text and figures:

Response: We greatly appreciate the encouraging comments and constructive suggestions from the referee. In the revised manuscript, we addressed all these concerns.

1. Are PXY and PXL2 also able to form complexes with SERK1 and BAK1, and are they able to bind CLE19?

Does SERK2 also function as a co-receptor with PXY/PXL1/PXL2?

Additionally, can PXL1 form homodimers or heterodimers with PXY and PXL2?

Although I don't think these experiments are absolutely required for publication, they would make the study more complete. Alternatively, greater speculation on how this complex might function would be desirable in the discussion.

Response: Thanks for the referee's comments. To address these questions, we performed both co-IP assay and gel-filtration assays and added these results to the revised manuscript (Fig. 5i-k, Fig. S8).

- 1) The ITC assay show that CLE19 directly interacts with PXL2 with a binding affinity of ~714nM, but has no interaction with PXY (Response Fig. 4a-b, see below, and Fig. 5i-j of the revised manuscript). Therefore, CLE19 can be perceived by both PXL1 and PXL2, but not PXY.*
- 2) The co-IP assay showed that CLE19 was also able to induce the interaction between PXL2 and SERK1 (Fig.5k) and between PXL2 and BAK1 (Fig. S8b). However, CLE19 was unable to induce the interaction between PXY and any of the SERKs (Response Fig. 4c-e, see below, and Fig. 5k and Fig. S8a-b of the revised manuscript).*

- 3) The Co-IP assay also showed that CLE19 can induce a strong interaction between SERK2 and PXL1, and a weak interaction between SERK2 and PXL2. However, CLE19 could not induce the interact between SERK2 and PXY ((Response Fig. 4d, below, and Fig. S8a of the revised manuscript).
- 4) Previous study has found that TDIF/CLE41 can induce SERK2-PXY interaction¹.
- 5) The gel filtration assay showed that the PXL1 is a monomer in solution (Fig. 5c of the revised manuscript), similar to the previous findings that PXY is a monomer in solution as well¹. It is possible that PXL1/PXL2/PXY form homodimers or heterodimers, but more analyses are still needed to answer this question.

Thus, we have added novel data to Fig. 5 and a supplemental figure (Fig. S8), and a discussion of a working model for in the revised manuscript as suggested. We have added a description of these results (Lines 310-335) and speculated on the possible working system in the discussion section (Lines 369-384) of the revised manuscript.

Response Fig. 4. (a-b) Measurements of binding affinity between CLE19 and PXL2 (i), and between CLE19 and PXY (j) by ITC. (c-e) In vivo co-IP assay using the *Nicotiana benthamiana* leaf transient expression system showing the interaction between PXL1 ΔKD /PXL1 ΔKD -R417419A/ PXL2 ΔKD / PXY ΔKD and SERK1 ΔKD (c)/ SERK2 ΔKD (d)/ BAK1 ΔKD (e) in the presence or absence of CLE19 treatment. (a-b) were shown

as Fig. 5i-j of the revised manuscript, (c) was shown as was shown as Fig. 5k, and (d-e) were shown as Fig. S8a-b in the revised manuscript.

Reference:

1. Zhang, H., Lin, X., Han, Z. et al. Crystal structure of PXY-TDIF complex reveals a conserved recognition mechanism among CLE peptide-receptor pairs. *Cell Res.* **26**, 543–555 (2016).

2. *For many of the experiments, the number of replicates performed is not clear. The number of needs to be explicitly stated in the manuscript, including in figure legends.*

Response: Thanks for pointing out this. The number of replicates is three in all the quantitative experiments; As suggested, this information that “Three biological replicates were performed. Data are shown as the mean ± SEM” has been stated in the figure legends of the revised manuscript.

3. *Some of the figures are missing the statistical tests needed to definitively conclude whether the phenotypes seen are meaningful. e.g. chi-squared tests or another appropriate statistical test for Figs 3E and 4D, S7D. For these sort of figures, it would also be normal to include how many pollen grains were phenotyped.*

Response: Thank the referee for this comment. As suggested, we conducted statistical tests for *Figs 3E and 4D, S7D* (Figs 2d and 2f, 3f-i, 4e-j, and S1b-c, S4b, S4d and S6b-e in the revised manuscript), and the pollen amount used in each sample was indicated.

4. *Line 86 – ‘CLE19 and other CLEs act as brakes to limit AMS expression’ – which other CLEs? Is it possible these CLEs are also bound by PXY/PXL1/PXL2? This could be added to the discussion.*

Response: Thanks very much for this valuable suggestion. Our previous studies supported the roles of CLE9, CLE16, CLE17, CLE41, CLE42, and CLE45 in pollen wall development by acting as negative regulators of AMS and downstream transcriptional pathways¹. In the assay of current study with CLV3, CLE3, CLE6, CLE8, CLE9, CLE11, CLE12, CLE14, CLE16, CLE18, CLE19, CLE20, CLE25, CLE40, CLE41/TDIF, CLE43, CLE45, and CLE46 (Supplemental Dataset 1), CLE19 was the only CLE peptide that co-purified with PXL1, indicating that CLE19 binds PXL1 with much higher binding affinity compared to other CLE peptides.

On the other hand, CLE41/TDIF has been shown to directly interact with PXY/PXL1/PXL2 by pull-down assays and ITC². Specifically, the binding affinity of CLE41/TDIF to PXL1 was 2.1 μM, and that of CLE41/TDIF to PXL2 was 9.9 μM², which are less than the binding affinities of CLE19 to PXL1 and PXL2 of ~346 nM and ~714 nM, respectively (Fig. 1b and 5i of the revised manuscript). These results suggested that the receptors PXL1 (and possibly PXL2) are more specific to the CLE19 peptide. We have added the above discussion in the revised text (please see Lines 369-384).

Reference:

1. Wang S., Lu J., Song X., et al., Cytological and transcriptomic analyses reveal important roles of CLE19 in pollen exine formation. *Plant Physiol.* 2017. **175**, 1186-1202 (2017).

2. Zhang, H., Lin, X., Han, Z. et al. Crystal structure of PXY-TDIF complex reveals a conserved recognition mechanism among CLE peptide-receptor pairs. *Cell Res.* **26**, 543–555 (2016).

5. *Line 120 – for readers less familiar with the field, it might be helpful to directly state why interactions were tested between CLE19 and only the LRR domains of PXL1.*

Response: Thanks for this suggestion. A previous study showed that the mature 12-aa CLE peptide bound to the extracellular LRR domain of PXL1¹; that is why we tested the interaction between CLE19 and the LRR domain of PXL1 (PXL1^{LRR}). We have stated this in the revised manuscript.

Reference:

1. Zhang, H., Lin, X., Han, Z. et al. Crystal structure of PXY-TDIF complex reveals a conserved recognition mechanism among CLE peptide-receptor pairs. *Cell Res.* **26**, 543–555 (2016).

6. *I would be hesitant about calling a peptide a hormone. A peptide signal might be a better way of phrasing this.*

Response: Thanks for this comment. We have rephrased it as suggested.

7. *Fig 1 panel A left diagram needs further explanation. What are the coloured shapes? Is this a schematic of the experiment, or actual results? I presume this is a schematic, however it needs further explanation in the figure legend.*

Response: Thanks for the comments. This diagram is a schematic diagram showing the identification of the peptide-receptor pair, and the coloured shapes presented a pool of peptides. In fact, this is a published method and a similar schematic diagram was also included in that paper¹. Therefore, we deleted this diagram to avoid distraction.

Reference:

1. Zhang, H., Lin, X., Han, Z. et al. Crystal structure of PXY-TDIF complex reveals a conserved recognition mechanism among CLE peptide-receptor pairs. *Cell Res.* **26**, 543–555 (2016).

8. *Fig 1 Panel B – it might be preferable if the legend refers to ‘PXL1 and CLE19’ rather than ‘PXL1-CLE19’ since a dash is otherwise used to indicate a fusion protein. Also, should specify plant species in the figure legend.*

Response: Thanks for this comment. We have changed it as suggested.

9. *Need to explain in text why CLE19G6T is assumed or known to be functionally inactive.*

Response: Thanks the referee for this comment. The function of the G6 amino acid in the CLE peptides was first demonstrated using an antagonistic peptide technology¹. These authors found the G6-to-T substitution of CLV3 gave the strongest antagonistic effects by exhibiting a dominant-negative clv3-like phenotype. They also demonstrated that such an antagonistic peptide technology is applicable to other CLV3/EMBRYO SURROUNDING REGION (CLE) peptides, CLE8 and CLE22, as well as in vitro treatments¹. In addition, previous studies in our own laboratory and other laboratory also revealed that transgenic plants with antagonistic CLE19_{G6T} construct that was expressed under the control of the CLE19 promoter exhibited dominant pollen developmental defects and a dominant seed abortion phenotype^{2,3}. All of these above results supported that the CLE19_{G6T} is inactive.

We have added the citation and explanation about the CLE19_{G6T} in the revised manuscript (please see Lines 82-82).

Reference:

1. Song X., Guo P., Ren S., et al., Antagonistic peptide technology for functional dissection of CLV3/ESR genes in Arabidopsis. *Plant Physiol.* **161**, 1076-1085 (2013).

2. Wang S., Lu J., Song X., et al., Cytological and transcriptomic analyses reveal important roles of CLE19 in pollen exine formation. *Plant Physiol.* **175**, 1186-1202 (2017).
3. Xu T., Ren S., Song X., and Liu C. CLE19 expressed in the embryo regulates both cotyledon establishment and endosperm development in *Arabidopsis*. *J. Exp. Bot.* **66**, 5367-5374 (2015).

10. Fig 1E and D – should label with size of proteins/ladder bands as an indicator of size.

Response: Thanks for the referee's comment. We labeled the size in the revised Fig. 1d-f.

11. Fig 1D - Need to state what the 0.57 is/how it has been calculated.

Response: Thanks for this suggestion. First, Image J software was used to measure the intensity of each band in Fig. 1d. Then, for the PXL1-FLAG bands, the intensity of the nonphosphorylated PXL1-FLAG band in the third lane from left to right was set to 1, and the value of the other bands was the ratio of their own intensity to that of this nonphosphorylated PXL1-FLAG band. Similarly, the intensity of HSP in the 3rd lane from left to right was set to 1, and the intensity of the other bands was expressed as a ratio to this band. As suggested, we have added the description in the revised figure legend.

"ImageJ software was used to measure the intensity of each band in Fig. 1d. After measurement, the intensity of the nonphosphorylated PXL1-FLAG band in the third lane from left to right was set to 1, and the value of the other bands was calculated as the ratio of their intensity to that of this nonphosphorylated PXL1-FLAG band. Similarly, the intensity of HSP in the third lane from left to right was set to 1, and the intensity of the other bands is presented as the ratio to this HSP band."

12. Fig 1E - Gel bands trending up for anti-FLAG and down for anti-HSP – this seems unlikely if both panels have been taken from the same gel. Could the authors please share an image of the entire blot with the reviewers?

Response: Thanks. Please see the entire blot images as following (also in the source data).

Response Fig. 5. *The original entire blot*

13. Fig 2 A/B – here it would be helpful to also show the brightfield image without the YFP overlay. It is very hard to see the cell boundaries in the merged image.

Response: Thanks for the referee's comment. We have provided the brightfield image as suggested (Response Fig. 6. /Fig. S3 of the revised manuscript).

Response Fig. 6. Subcellular localization of PXL1-YFP, PXL2-YFP, and PXY-YFP in *Nicotiana benthamiana* leaf transient expression system. Shown as Fig. S3 of the revised manuscript.

14. Fig S1C – I don't find this image convincing – it is very hard to make out GUS staining on the image.

Response: Thanks for this comment. We agree with the referee that the GUS signal in Fig S1C is weak. This was because to obtain clear cell layer information, we performed a semithin sectioning of the anther, and the image was taken from the semithin section. In the revision, we have collected new data with higher quality (Response Fig. 7/ Fig. 2g in the revised manuscript).

Response Fig. 7. Expression of PXL1, PXL2 and PXY genes in inflorescences, anthers and anther lobes using PXL1::GUS, PXL2::GUS and PXY::GUS reporter lines. bar =100 μ m for inflorescences, 50 μ m for anthers, 10 μ m for anther lobes. The expression in early floral buds is indicated with red arrowhead, and the anther tapetum layer in each lobe is indicated by dotted lines. Shown as Fig. 2g in the revised manuscript

15. Fig 2F – while the qPCR shows low expression of PXL1 in the *pxl1* T-DNA insertion lines, these primers span the T-DNA insertion site. To show that there are low levels of transcript throughout and therefore that a truncated protein is unlikely to be produced, primers that are 5' of the insertion site should also be used for qPCR.

Response: Thank the referee for this comment. As suggested, we used another pair of primers located 5' to the insertion site to examine the expression level of the gene in the corresponding T-DNA insertion lines (Figure S1 and S4 in the revised manuscript). Amplification with primers that are 5' to the insertion site showed that there is some expressions of the trunked PXL1 mRNA in *pxl1-1* and *pxl1-2* (Figure S1), a low level of PXL2 mRNA in *pxl2-1*, and the truncated PXY transcript in *pxy-3* (Figure S4).

16. Fig 3B, F and G (and others) SD or SEM on qPCRs seem surprisingly small for biological samples. Are these calculated on technical replicates rather than biological replicates? Also need to specify whether SEM or SD is shown for.

Response: Thanks for the referee's comment. All these calculations are based on the SEM of 3 technical replicates of each WT or homozygous transgenic plants, and SEM is used for this calculation. This information has been added to the figure legend.

17. Fig 3E – it is incorrect to describe this as 'statistical results'. Proportion of pollen grains by phenotype might be a better way to express this.

Response: Thanks for the referee's comment. As suggested, we have corrected the description as "Proportion of pollen with normal, moderate-D and severe-D exine defects from various genotypes." In the figure legends.

18. Fig S6 – please add in qPCR primer locations.

Response: Thanks for the referee's comment. We have added the qPCR primer locations in the revised Figure S1 and Figure S4.

19. Lines 234 to 238 'To test this idea, *pxl2-1*, *pxy-3*, and *pxy-5* single knockout mutants (Fig. S6) were analyzed for their pollen developmental phenotypes. Anthers from all three single mutants were slightly smaller, and their pollen counts in each anther was obviously reduced as well, similar to those of *pxl1-1* and *pxl1-2*.' This data is not shown in Fig S6, nor could I find it elsewhere in the paper – this should be added to support the statement.

Response: Thanks for the referee's comment. We have added the information of pollen number in *pxl2-1*, *pxy-3* single knockout mutants, and statistical analysis in the revised Figure 2d.

20. Fig 4 – figure legend describes panels A and B in opposite order to what is shown.

Response: Thanks for the referee's comment. The error has been corrected.

21. Arrows, asterisks, boxes are often shown on figures with no additional explanation of what they mark given in the figure legend. In Fig S7A, arrow heads are VERY small and hard to see.

Response: Thanks for the referee's comment. We have added the detailed explanation of the asterisks, boxes, and magnified the arrow head in the revised Figure 2 and Figure legend.

22. Line 250 – What is known of MOL1's function and why is it of particular note in the phylogenetic tree in Fig S8? This should be added to the text.

Response: Thanks for the referee's comment. MOL1 is known to be required for cambium homeostasis in *Arabidopsis thaliana*. However, there is no evidence that MOL1 is able to recognize CLE peptides. It is also not related to our study. Therefore, we decided to remove the information about MOL1 from the text to avoid distraction.

23. Line 255 – amino acid similarity between PXL1 and PXL2 is commented on. What of PXY compared to PXL1 and PXL2?

Response: Thanks for the referee's comment. As suggested, we have added this information in the revised manuscript (Lines 151-152) as "PXL1 and PXL2 show 50.54.% sequence identity at the amino acid level, whereas the sequence identity between PXY and PXL1 and that between PXY and PXL2 are 29.25% and 27.44%, respectively".

24. Line 261 – It would be helpful to remind the reader what the developmental defects of the CLE19-OE line are.

Response: Thanks for the referee's comment. The information of CLE19-OE defects has been described in Lines 248-251 as following:

"In comparison, only 68% and 60% of the pollen in C#11 and C#18 plants was normal, and 32% and 30% showed pollen wall defects. Specifically, 26% and 17% of pollen showed severe-C defects, and 6% and 24% showed moderate-C defects, in C#11 and C#18 plants, respectively (Fig. 4d-f)."

The C#11 and C#18 here are two independent CLE19-OX transgenic lines, which has been described in Lines 233-235 as "three new homozygous CLE19-OX lines (C#11, C#16 and C#18) and two homozygous CLE19-OX/DN-PXL1 double transgenic lines (CDM#3 and CDM#4) were chosen for further phenotypic analyses."

25. Line 302 – I don't understand what is meant here '....', indicating that the CLE19 signal and CLE19

signal through PXL1 are (at least partially) dependent on the presence of a functional PXL1 receptor.' It might be advisable to shorten the sentence and make the meaning clearer.

Response: Thanks for the referee's comment. We have rephrased the sentence to "..., indicating that CLE19 signal transduction is PXL1 dependent".

26. Line 320 – '*serk1+/- serk2-/-bak1-/-*' Please be consistent with how you refer to this line in the manuscript.

Response: Thanks for the referee's comment. We refer it as *serks* multiple mutants in the revised manuscript.

27. Line 320 – *I believe Fig S9A is the correct figure to be cited here. Also, while serk1 is referred to in the text as having been pollen phenotyped, this is not included in the figure. For completeness, it should be included.*

Response: Thanks for the referee's comment. The citation error has been corrected. The *serk1* mutant data has been added to the revised Figure S6.

28. Fig 5 – *order of panels could be improved.*

Response: Thanks for the referee's comment. The panels of Figure 5 have been reordered.

29. Line 401 – *I think it would be more correct to say that a peptide is 15ecognized by the receptor than vice versa, or to say that the peptide can bind to different receptors.*

Response: Thanks for the referee's comment. The sentence has been rephrased.

30. *What about pollen phenotypes of PXY/PXL double mutants? Are these intermediate to single and triple phenotypes?*

Response: Thanks for the referee's comment. The double mutants *pxl1pxl2*, and *pxl2 pxy* showed intermediated phenotypes to single and triple mutants (Please see the Figure below).

Response Fig. 8.

31. Light levels and type of illumination should be included in the description of the plant growth conditions, along with type of soil used.

Response: Thanks for the referee's comment. This part had been added to the revised method and materials part (Line 471 -Line 473).

32. pGWB vectors – don't these usually have three digit codes where the first number indicated the selection marker? Please include the full number.

Response: Thanks for pointing out this. The full number has been added (Line 478).

33. In figure legends, please be clear on which experiments were performed in Arabidopsis and which in tobacco.

Response: Thanks for the referee's comment. We have added detailed plant types used in corresponding experiments into the revised figure legends.

34. Please specify Agrobacterium strain used for all plasmids transformed into Arabidopsis or tobacco.

Response: Thanks for the referee's comment. The information of Agrobacterium strain GV3101 has been added to the revised method.

35. Line 486-489 'To analyze the expression level of PXL1, PXL2 and PXY in the corresponding T-

DNA mutants, PXL1LRR level in DN-PXL1 transgenic plant, tapetum development and pollen exine formation related genes expression level in DN-PXL1 and CLE19-OX/DN-PXL1 transgenic plants, CLE19 expression level in transgenic mutants.’ I am unclear on your meaning here. Please rephrase.

Response: Thanks for the referee’s comment. We have rephrased the description in lines 467- 470 of the revised manuscript as follows: “We used qRT–PCR to analyse the expression levels of PXL1, PXL2 and PXY in the corresponding T-DNA mutants, the PXL1^{LRR} expression level in DN-PXL1 transgenic plants, the expression levels of genes related to tapetum development and pollen exine formation in DN-PXL1 and CLE19-OX/DN-PXL1 transgenic plants, and the CLE19 expression level in transgenic mutants.”.

36. There are a few grammatical and spelling errors that could do with correction. This mostly affects the materials and methods and figure legends, however there are also a few errors elsewhere in the manuscript.

Response: Thanks. The errors have been corrected in the revised manuscript.

Reviewer #3 (Remarks to the Author):

Peptide hormones are important factors involved in diverse developmental and physiological processes. In this study, the authors searched the cognate receptors that recognize CLE19 peptide by a library-based interaction ability screening. The primary findings of this study are the identification of the PXL1 and related receptors as well as co-receptors for CLE19 peptide, and the module plays a role in the regulation of pollen development. The research plans, especially genetic analyses seem well organized and the results will be of interest to readers interested in plant physiology and small molecule signaling peptides. I very much enjoyed reading this manuscript and I think that I can recommend this for publication from Nature communications.

However, the current version manuscript contains several concerns and needs to be improved before acceptance. Most importantly, many of the WB photos are of poor quality hence the results are not convincing. Particularly, in Fig. 1D, the important phosphorylated PXL1 signal is not very clear, similarly, in Fig 1E, the loading control varies greatly. The authors should consider replacing the pictures with ones that better depict the results.

Response: Thanks for the constructive and valuable comments from this referee. We followed the referee’s suggestions to further improve our manuscript by providing photos with high quality. We have optimized the experiments and achieved WB photos with much higher quality. All these pictures have been replaced (Fig. 1d-f).

Response Fig. 9. (the same as the revised Fig. 1d-f.) (d) Seedlings of pPXL1::PXL1-FLAG transgenic plants were treated with or without CLE19 peptide. A band correlated with PXL1-FLAG (indicated by red arrowhead) was shifted after incubation with CLE19 for 1.5 hours in phos-tag gel, which disappeared after

App treatment. HSP was used to indicate the input amount. (e) Seedlings of *pPXL1::PXL1-FLAG* transgenic plants were treated with 20 μM CLE19 for 1.5 hours, then PXL1-FLAG proteins were pulled down by FLAG beads. The CLE19-induced phosphorylation of PXL1 was verified by pT/S anti-body, and the anti-FLAG antibody was used to indicate the input. (f) CLE19_{G6T} or CLV3 did not induce the phosphorylation-related migration of PXL1. The phosphorylated bands are indicated by red arrows, and the unphosphorylated bands are indicated with blue arrowheads in (d-f).

Next, the authors claim that PXL1 is the cognate receptor for CLE19, which has already been reported as the receptor for TDIF. Genetic data shown in this manuscript adequately support this hypothesis, but there is insufficient discussion of the similarities and differences between CLE19 and TDIF.

In this context, showing sequence alignments and comparison of the peptide hormones, namely CLE19 and TDIF will be valuable.

In addition, since the crystal structure of TDIF-PXY has already been reported, it is desirable to make a comparison with it. Specifically, comments should be made regarding the amino acid residues responsible for the interaction. Introducing mutation(s) to the interaction domain of PXY for the TDIF binding residues should also be tested for the ability to interact with CLE19. Is the position of the DN mutation in CLE19 explainable to the opposing effects?

Response: Thanks for this important comment. Yes, PXL1 has been reported as a receptor of TDIF previously¹. In that study, TDIF directly interacts with PXY/PXL1/PXL2 with affinities of, respectively, 33 nM, 2.1 μM and 9.9 μM ¹. In our study, the binding affinities of the CLE19 to PXL1 and PXL2 were ~346 nM and ~714 nM, respectively (Fig. 1b and 5i), suggesting that the receptors PXL1 and PXL2 are more specific to the CLE19 peptide.

As suggested, we now showed sequence alignments and comparison of CLE19 and TDIF (Fig. 5g) in the revised manuscript. The amino acid identity between CLE19 and TDIF is 50%.

In previous TDIF-PXY crystal structure, the RxR motif of PXY was shown critical for TDIF peptide binding; the RxR motif is also conserved in several other small peptide-receptor interactions. Thus, we synthesized the PXL1 protein with the R417A and R419A double substitutions to test whether the RxR motif is important for CLE19-induced interaction between the PXL1 receptor and its SERK coreceptors. As expected, the co-IP assay showed that PXL1^{R417A/R419A} could not interact with SERK1/BAK1, regardless of the presence of CLE19 in the reaction (revised Figure 5k), suggesting that the CLE19-PXL1 interaction is analogous to the TDIF-PXY interaction and is a prerequisite for the interaction between PXL1 and its SERK coreceptor.

According to the TDIF-PXY complex structure, the G6 site of CLE has no direct interaction with the receptor LRR domain, suggesting that the G6T mutation might not destroy the interaction between the ligand–receptor pair. However, our phosphorylation results revealed that CLE19_{G6T} could not induce PXL1 phosphorylation as the WT CLE19 did (Fig. 1f), suggesting that G6 has a role separate from the interaction of CLE19 with PXL1. We propose that the fact that the CLE19_{G6T} mutant peptide still has PXL1-interaction capability but lost the ability of inducing PXL1 phosphorylation might explain why CLE19_{G6T} showed opposing effects to the WT CLE19.

We have added novel discussion in the revised text (please see Lines 368-385).

Fig. 1b**Fig. 5i****Fig. 5g**
Response Fig. 10. The revised Fig. 1b, 5i and 5g. (Fig. 1b) Measurement of binding affinity between CLE19 and PXL1^{LRR} by ITC. Top panel: twenty injections of CLE19 solution were titrated into PXL1^{LRR} solution in the ITC cell. The area of each injection peak corresponds to the total heat released for that injection. Bottom panel: the binding isoform for CLE19 and PXL1^{LRR} interaction, the integrated heat is plotted against the molar ratio between CLE19 and PXL1^{LRR}. Data fitting revealed a binding affinity of about 346 nM. (Fig. 5i) Measurement of binding affinity between CLE19 and PXL2. (Fig. 5g) Sequence and structure comparison between CLE19 and TDIF.

Reference:

1. Zhang, H., Lin, X., Han, Z. et al. Crystal structure of PXY-TDIF complex reveals a conserved recognition mechanism among CLE peptide-receptor pairs. *Cell Res.* **26**, 543–555 (2016).

For SERKs, the main argument of the authors that SERKs are acting as co-receptor for CLE19-PXL1 is based on the mutant phenotypes as well as biochemical experiments. Although the genetics provided in this manuscript is very sophisticated, the epistasis between SERKs and CLE19 or PXY is not fully proved. Although I think the regulatory mechanisms that involve the CLE19-PXY-SERK module is a promising hypothesis, evidence by genetic analysis of SERKs and CLE19 or PXY is desirable to establish a convincing model.

Response: Thanks for the referee's comments. SERKs have been reported as coreceptors for several LRR XI family RLKs. In particular, SERKs have been shown as coreceptors for TDIF-PXY signalling¹. In this study, using the in vitro gel-filtration and in vivo co-IP assays, we showed that interactions between PXL1 and SERKs were induced by CLE19 (Fig 5b-e). In addition, we also examined expression of genes for the tapetum and pollen wall development in the *serk1*+/- *serk2* *bak1* triple mutant. The results showed that the expression of *AMS*, *MS1* and the pollen exine formation genes *CYP98A8*, *ACOS5* also was increased in the triple mutant (Figure S6d-e), similar to the increased expression in the *DN-PXL1* transgenic plants; these results support the hypothesis that SERKs function as coreceptors of PXL1 for CLE19. Due to the fact that the *serk1 serk2* double mutant is male sterile^{2,3}, it is impractical to examine CLE19-PXL1 interaction in the *serk1 serk2* double mutant background. We hope that our current in vitro and in vivo biochemical data, pollen phenotypes and downstream gene expression change can support the model that SERKs act as coreceptors with PXL1 for recognition of the CLE19 signal.

Reference:

1. Zhang, H., Lin, X., Han, Z. et al. Crystal structure of PXY-TDIF complex reveals a conserved recognition mechanism among CLE peptide-receptor pairs. *Cell Res.* **26**, 543–555 (2016).
2. Albrecht, C., Russinova, E., Hecht, V., Baaijens, E., and de Vries, S. The Arabidopsis thaliana SOMATIC EMBRYOGENESIS RECEPTOR-LIKE KINASES1 and 2 control male sporogenesis. *Plant Cell.* **17**, 3337–3349 (2005).

3. Colcombet, J., Boisson-Dernier, A., Ros-Palau, R., Vera, C.E., and Schroeder, J.I. Arabidopsis SOMATIC EMBRYOGENESIS RECEPTOR KINASES1 and 2 are essential for tapetum development and microspore maturation. *Plant Cell*. 17, 3350–3361 (2005).

For RT-PCR analyses of pollen exine formation genes, as the context of tissue-specificity is important to this study, analysis by bulk RNA alone is not sufficient. It is desirable to observe the markers for (selected) genes.

Response: Thanks for the referee's comment. As suggested, we have examined the expression of *DYT1*, *AMS*, and *MS1*, which are known as key regulatory genes for tapetum and pollen development¹. We examined the expression variation of these genes in different genotypes to help understand the function of PXL1 in pollen and anther development.

Reference:

1. Ma, H. Molecular genetic analyses of microsporogenesis and microgametogenesis in flowering plants. *Annu. Rev. Plant Biol.* 56, 393-434 (2005).

It will be interesting to discuss whether CLE19 binding and TDIF binding are similar or different stimuli for receptors. This is entirely optional, function replacement experiments with CLE19p:TDIF and/or reciprocal replacement may help this discussion.

Response: Thanks for this valuable suggestion. We have added structural modeling of CLE19–PXL1 (Fig. 5f-h) and a discussion of the similarity of the CLE19–PXL1 and TDIF–PXY interactions in the revised manuscript. (Lines 325-330 and Lines 368-385).

Fig. 5f-h. Structure of CLE19-PXL1^{LRR}/SERK1^{LRR} complex.

(f) The overall structure of CLE19-PXL1^{LRR}/SERK1^{LRR} complex. (g) Structure comparison between CLE19 and TDIF. (h) Detailed interactions of the boxed region in (f). The side chain of the last residue from CLE19 and the R417/R419 sites from PXL1 surface are labeled. (i-j) Measurement of binding affinity between CLE19 and PXL2 (i), and between CLE19 and PXY (j) by ITC. (k) *In vivo* co-IP assay using *Nicotiana benthamiana* leaf transient expression system showing the interaction between PXL1^{ΔKD}/PXL1^{ΔKD-R417/419A}/PXL2^{ΔKD}/PXY^{ΔKD} and SERK1^{ΔKD} with or without CLE19 treatment.

minor
L338

"interaction

point
signal"

I think the word is inappropriate (overstatement or misleading). The signal itself indicates the MYC signal or PXL1-MYC, and the presence of the signal suggests the interaction.

Response: Thanks for this comment. The “interaction signal” has been revised to “interaction”.

typos

L338 CLE19treatment need space

Response: Thanks for pointing out this. The space has been added.

Reviewer #1 (Remarks to the Author):

I do appreciate all the work the authors have done following up on the reviewers' concerns. However, there are some minor issues that still need to be addressed before publication.

Figure 1d: The authors said that "the intensity of the non-phosphorylated PXL1-FLAG band in the third lane from left to right was set to 1, and the value of the other bands was calculated as the ratio of their intensity to that of this non-phosphorylated PXL1-FLAG band". However, it is not clear to me how the total intensity of the band in the last lane (+CLE19, + peptide, + λ pp) is 1 when it should be the sum of the intensities of both bands (1 and 0.5) of the lane "+CLE19, + peptide", that is 1.5.

Figures 2e/f: I guess that the colors used to define the exine lessons are wrong. According to figure 1f, grey should be normal; blue, moderate and green, severe. The same for figure 3g. In figure 4d is correct.

In the statistical tests of Figures 2 and 3, "n" must be the number of independent experiments, not the number of anthers counted.

In Figures 2f and 3g, please indicate whether the statistical tests were performed between normal and the sum of moderate-D and severe-D.

Reviewer #2 (Remarks to the Author):

This is a revised manuscript from Yu and colleagues identifying PXL1 as a receptor for the CLE19 peptide. The authors have now addressed comments from three reviewers. All my concerns have now been addressed.

My one remaining concern is regarding the statistics applied throughout the study.

My original comment was: Fig 3B, F and G (and others) SD or SEM on qPCRs seem surprisingly small for biological samples. Are these calculated on technical replicates rather than biological replicates? Also need to specify whether SEM or SD is shown for.

Response: Thanks for the referee's comment. All these calculations are based on the SEM of 3 technical replicates of each WT or homozygous transgenic plants, and SEM is used for this calculation. This information has been added to the figure legend.

Here the authors seem to be saying the SEM has been calculated based on technical replicates. SEM should always be calculated based on biological replicates (but using the mean for each biological replicate as derived from technical replicates) as it is the biological variation which is important in determining if two samples are statistically the same or different. The correct statistics need to be presented for each figure, and ideally where practical individual data points should be shown for each biological replicate rather than just a single bar showing the mean of the biological replicates.

Reviewer #3 (Remarks to the Author):

Although the authors responded well to reviewers' concerns, I think several changes are still required before acceptance.

Figure1b
Indication of Y-axis in lower column is incorrect (-12.0)

Figure 1e

WD image for anti-p is not clear, the authors are preferred to replace this with a clearer one.

Figure 2d

Please provide the numbers in Y-axis in a line.

Figure 2f

In Y-axis, "%" is overlapped to scale line.

Figure 2e/f

Indicative legends to Moderate-D/Normal/Severe-D in 2e, probably for 2f? And that seems incorrectly in the color, green for Moderate/Severe...

Figure 2h legend

Images for benthamiana leaf cells are not included? Probably in S3.

L214-219, Figure 3f/g

In this section, comparisons between DM#8 and DM#8 pxl1-1, DE#12 and DE#12 pxl1-1 are included, as well as the comparison with WT. Please provide discussions with multiple comparison statistics.

Figure 3f

The indication of "20%" is partly behind a white line?

L228 PXL1 is required for the function of CLE19

Since the introduction of pxl1-1/- into DE#12 (fig3f) exhibited an additional effect, the expression of DN-PXL1 did not abolish endogenous PXL1 function completely. Therefore, I think the CDM#3 and CDM#4 transgenic lines still harbor functional PXL1, which did not convince the authors' claim that "PXL1 is required for the function of CLE19". In that case, the introduction of the loss-of-function of PXL1 (e.g. pxl1-1/-) into the CLE19-OX and examining whether the mutation attenuates the effect of CLE19-OX or not seems more appropriate than the use of DN-PXL1. Simultaneously, evaluating the effect of loss-of-PXL2 or PXY to ask whether the mutations attenuate the abnormalities of CLE19-OX or not also adds molecular and physiological insights into the redundancy and differences of the RLKs.

Figure S5b "Relative expression level PXL1LRR)"

Bracket is disappeared.

Figure S5 PXL1FL

I think the description "FL" is confusing. The RT-PCR targeted Kinase Domain, rather than Full Length, the use of KD is preferable.

Figure S5 legend

(c) F12/R2 : typo

Figure 5i/j

Please check the indications of the Y-axis in these graphs. I think these scale lines seem inconsistent, showing in two lines in (upper in i). They make the graphs difficult to understand.

Fig S8 anti-MYC bands

According to the Input WB, PXL1KD-MYC and PXL2KD-MYC were detected at similar sizes, conversely, these proteins seem to be detected as different sizes in the IPed WD. The authors are preferred to explain this. If this matter comes from the winding in gel running, the authors are preferred to replace the images with ones that better depict the results.

The construction detail and plant materials for protoplast assays should be provided.

Please re-check the overall statistics included in this manuscript. Especially, the use of p-value is

inconsistent through the manuscript, namely the authors sometimes use "p=" and "p<" in a figure e.g. Fig3h/I". These are very confusing.

Point-to-point Responses to Referees

Reviewer #1 (Remarks to the Author):

I do appreciate all the work the authors have done following up on the reviewers' concerns. However, there are some minor issues that still need to be addressed before publication.

Figure 1d: The authors said that “the intensity of the non-phosphorylated PXL1-FLAG band in the third lane from left to right was set to 1, and the value of the other bands was calculated as the ratio of their intensity to that of this non-phosphorylated PXL1-FLAG band”. However, it is not clear to me how the total intensity of the band in the last lane (+CLE19, + peptide, + λ pp) is 1 when it should be the sum of the intensities of both bands (1 and 0.5) of the lane “+CLE19, + peptide”, that is 1.5.

Response: Thank the referee for your astute observation and constructive comment. In the previous version, we hypothesized that the software ImageJ may not have accurately quantified the total protein amount in the electrophoresis gel due to excessive protein loading in the third and fourth lanes. In the current revised version, we reduced the protein loading in each lane, conducted a re-Western blot, and quantified the electrophoretic bands after exposure. We set the intensity of the non-phosphorylated PXL1-FLAG band in the first lane as a reference value of 1, and the value of each detected band in the second, third and fourth lanes as a ratio of the intensity of the reference band. Our results showed that in the third lane (+CLE19), 0.5 of the PXL1 protein was phosphorylated and migrated, while 0.9 of PXL1 protein exhibited non-phosphorylated electrophoretic behavior. For the fourth lane (+CLE19, + λ pp), all 1.4 of PXL1 protein were non-phosphorylated, and no migrated band was observed. These results suggest that CLE19 promotes the phosphorylation of PXL1 in plants, which could be reversed by λ pp. Please refer to Response Fig. 1.

Response Fig. 1 (Fig. 1d of current manuscript).

Seedlings of *pPXL1::PXL1-FLAG* transgenic plants were subjected to CLE19 treatment or left untreated. Electrophoretic mobility of the phosphorylated PXL1-FLAG band (indicated by a red arrowhead) was altered in the phos-tag gel after a 1.5-hour incubation with CLE19, which was abolished by treated with λ pp. HSP was used to indicate the input amount. The intensity of each band in Fig. 1d was measured using ImageJ software. After measurement, the intensity of the non-phosphorylated PXL1-FLAG band in the first lane from left to right was set as a reference value of 1, and the value of each detected band in the second, third and fourth lanes were expressed as the ratio of the detected band's to that of the reference band. Similarly, the intensity of HSP in the first lane from left to right was set to 1, and the intensity of other bands was presented as the ratio to this HSP band.

Figures 2e/f: I guess that the colors used to define the exine lessons are wrong. According to figure 1f, grey should be normal; blue, moderate and green, severe. The same for figure 3g. In figure 4d is correct.

Response: Thanks for pointing this out. These mistakes have been revised in the current manuscript.

In the statistical tests of Figures 2 and 3, “n” must be the number of independent experiments, not the number of anthers counted.

Response: Thank the referee for this comment. In the current manuscript, we conducted three independent trials, each counting 10 anthers, and re-completed the statistical tests using data from these three independent trials (Response Fig.2 and 3).

Response Fig. 2 Quantification of pollen amounts per anther in the WT and *px1*, *px2* and *pxy* single, double and triple mutants. (a) Fig. 2d of the current manuscript. Quantification of pollen amounts per anther in indicated genotypes. Three individual biological repeats (three independent experiments) were performed. Different letters represent significant difference between each other, $P < 0.05$, one-way ANOVA with Tukey multiple comparison test. (b-d) Quantification results of each biological repeat (independent experiment) that used for (a). Ten anthers were used for each biological replicate.

Response Fig. 3 Quantification of pollen amounts per anther in the WT and *DN-CLE19*, *DN-PXL1-Myc*, *DN-PXL1-EYFP*, *DN-PXL1-Myc px1+/-* and *DN-PXL1-EYFP px1-/-* transgenic plants. (a) Fig. 3f of the current manuscript. Quantification of pollen amounts per anther in indicated genotypes. Three individual biological repeats (three independent experiments) were performed. Different letters represent significant difference between each other, $P < 0.05$, one-way ANOVA with Tukey multiple comparison test. (b-d) Quantification results of each biological repeat (independent experiment) that used for (a). Ten anthers were used for each biological replicate.

In Figures 2f and 3g, please indicate whether the statistical tests were performed between normal and the sum of moderate-D and severe-D.

Response: Thanks for this suggestion. The statistical tests were performed between normal and the sum of moderate-D and severe-D. This information has been added into the figure legend.

Reviewer #2 (Remarks to the Author):

This is a revised manuscript from Yu and colleagues identifying PXL1 as a receptor for the CLE19 peptide. The authors have now addressed comments from three reviewers. All my concerns have now been addressed.

My one remaining concern is regarding the statistics applied throughout the study.

My original comment was: Fig 3B, F and G (and others) SD or SEM on qPCRs seem surprisingly small for biological samples. Are these calculated on technical replicates rather than biological replicates? Also need to specify whether SEM or SD is shown for.

Response: Thanks for the referee's comment. All these calculations are based on the SEM of 3 technical replicates of each WT or homozygous transgenic plants, and SEM is used for this calculation. This information has been added to the figure legend.

Here the authors seem to be saying the SEM has been calculated based on technical replicates. SEM should always be calculated based on biological replicates (but using the mean for each biological replicate as derived from technical replicates) as it is the biological variation which is important in determining if two samples are statistically the same or different. The correct statistics need to be presented for each figure, and ideally where practical individual data points should be shown for each biological replicate rather than just a single bar showing the mean of the biological replicates.

Response: Thanks the referee to point this out. Yes, the original calculations were based on three technical replicates. As you suggested, in the current manuscript, we have performed three biological replicates for the qPCR analyses and have presented the correct statistics for Fig. 3B (current Fig. S5b-c), Fig. 3F and G (current Fig. 3h and i). Accordingly, we have revised the figures to show the practical individual data points for each biological replicate.

Reviewer #3 (Remarks to the Author):

Although the authors responded well to reviewers' concerns, I think several changes are still required before acceptance.

Figure 1b. Indication of Y-axis in lower column is incorrect (-12.0)

Response: Thank the referee for pointing this out. We have corrected this in the current Fig. 1b.

Figure 1e. WD image for anti-p is not clear, the authors are preferred to replace this with a clearer one.

Response: Thank the referee for this comment. We have optimized the Co-IP and Western-blot experiments and achieved images with quality high enough (Response Fig. 4). Fig. 1e for anti-p have been replaced in the current manuscript.

Response Fig. 4 Current Fig. 1e.

Figure 2d. Please provide the numbers in Y-axis in a line.

Response: Thanks. We have corrected this in the current Fig. 2d.

Figure 2f. In Y-axis, "%" is overlapped to scale line.

Response: Thanks. We have corrected this in the current Fig. 2f.

Figure 2e/f. Indicative legends to Moderate-D/Normal/Severe-D in 2e, probably for 2f? And that seems incorrectly in the color, green for Moderate/Severe...

Response: Thanks for pointing this out. The legends for figure 2f have been corrected.

Figure 2h legend. Images for benthamiana leaf cells are not included? Probably in S3.

Response: Thanks. We have removed the description for benthamiana in the figures 2h legend.

L214-219, Figure 3f/g. In this section, comparisons between DM#8 and DM#8 pxl1-1, DE#12 and DE#12 pxl1-1 are included, as well as the comparison with WT. Please provide discussions with multiple comparison statistics.

Response: Thank the referee for this suggestion. We re-analysis the data for current figure 3f/g with multiple comparison statistics, as shown in the Response Fig. 5a-b. Letters were assigned by ANOVA and Tukey's multiple comparison test for Fig. 3f and by chi-square test for Fig. 3g. In consistent, we also re-analysis the data for current figure 2d and 2f with multiple comparison statistics, as shown in the response Fig. 5c-d.

Response Fig. 5 Phenotypic analyses with multiple comparison statistics.

(a and c) Statistical analysis results show the pollen amounts per anther of various genotypes. Letters were assigned by ANOVA and Tukey's multiple comparison test. Three biological replicates were performed. For each replicate, ten anthers were used for analysis. Every dot shows the average value for each biological replicate. (b and d) Proportion of pollen with normal (grey), moderate-D (blue) and severe-D (green) exine defects from various genotypes. Letters were assayed based on calculation by chi-square, and n indicates the number of anthers counted for each genotype. (a) Fig. 3f, (b) Fig. 3g, (c) Fig. 2d and (d) Fig. 2f for the current manuscript.

Figure 3f, The indication of "20%" is partly behind a white line?

Response: Thanks for pointing this out. The white line has been deleted.

L228 PXL1 is required for the function of CLE19

Since the introduction of *pxl1-1/-* into DE#12 (fig3f) exhibited an additional effect, the expression of DN-PXL1 did not abolish endogenous PXL1 function completely. Therefore, I think the CDM#3 and CDM#4 transgenic lines still harbor functional PXL1, which did not convince the authors' claim that "PXL1 is required for the function of CLE19". In that case, the introduction of the loss-of-function of PXL1 (e.g. *pxl1-1/-*) into the CLE19-OX and examining whether the mutation attenuates the effect of CLE19-OX or not seems more appropriate than the use of DN-PXL1.

Simultaneously, evaluating the effect of loss-of-PXL2 or PXY to ask whether the mutations attenuate the abnormalities of CLE19-OX or not also adds molecular and physiological insights into the redundancy and differences of the RLKs.

Response: Thank the referee for this insightful comment. Our experiments have demonstrated that the introduction of *pxl1* mutant into the DE#12 and DM#8 transgenic plants (Col-0 background) has increased the proportion of abnormal pollen walls (Fig. 3g), indicating that DN-PXL1 did not completely eliminate the function of PXL1 in the WT, and the presence of two copies of PXL1 protein in the DN-PXL1-MYC (DMs) and DN-PXL1-EYFP (DEs) transgenic plants still retained some function. Despite this, our results from CDM#3 (CLE19-OX#3 in DM#8 background) and CDM#4 (CLE19-OX#4 in DM#8 background) support the conclusion that "PXL1 is required for the function of CLE19". This conclusion is supported by a combination of genetic, biochemical, and gene expression obtained from our study. The reasons are as follows:

1. The use of dominant-negative (DN) strategy that involves truncated receptor kinases (Δ Kinase forms) has become a widely recognized and powerful approach to investigate biological functions and signaling patterns of receptor kinases in both animal and plant systems. This approach

is particularly useful when a single loss-of-function mutation may not produce significant phenotypic effects due to extensive functional redundancy. For instance, this approach has been effectively employed in elucidating the *in vivo* functions and signaling components of receptor tyrosine kinases such as fibroblast growth factor RTK and activin (transforming growth factor β) receptor Ser/Thr kinase in mesoderm formation during *Xenopus* embryogenesis (Amaya et al., 1991; Hemmati-Brivanlou and Melton, 1992). The application of truncated epidermal growth factor RTK in *Drosophila* eyes has also revealed the requirement of this RTK in the process of differentiation (Freeman, 1996). Moreover, the expression of truncated ERECTA protein lacking the cytoplasmic kinase domain (Δ Kinase) has been shown to confer dominant-negative effects, revealing redundancy in the Arabidopsis ERECTA leucine-rich repeat receptor-like kinase signaling pathway that regulates organ shape (Shpak et al., 2003).

2. The findings that DM#8 and DE#12 exhibited a more severe abnormality in pollen exine phenotype than the *pxl1* single and *pxl1 pxl2 pxy* triple mutants (Fig. 2f and 3g), along with the enhanced abnormal pollen exine phenotype upon the introduction of the *pxl1* mutant into DM#8 further enhanced the abnormal pollen exine phenotype (Fig. 3g), provide compelling evidence for the redundancy of the PXL1 signaling pathway. Similar observations have been documented in the case of DN-ERECTA (Shpak et al., 2003), where transgenic plants expressing the Δ Kinase-form of ERECTA (DN-ERECTA) display the *erecta-105* phenotype, but the DN-ERECTA *erecta-105* plants exhibit more severe growth defects than both DN-ERECTA and *erecta-105* mutant (Shpak et al., 2003). These observed phenotypic effects in both cases suggest that the signaling pathway controlled by these receptors are partially redundant and that multiple receptors are involved in the same downstream processes.

3. In this study, it was observed that PXL1, PXL2, and PXY function redundantly in the regulation of pollen exine development. The single mutants of *pxl1*, *pxl2*, and *pxy* exhibited mild abnormalities in pollen exine, while the phenotype became more severe in *pxl1 pxl2* double and *pxl1 pxl2 pxy* triple mutants (Fig. 2a and f). The transgenic plants carrying the DN-PXL1 construct (DM#8, DM#25, DE#12, and DE#20) displayed an overfilling phenotype of abnormal pollen exine similar to that of the *pxl*-related mutants, albeit with greater severity (Fig. 3a-g). This phenomenon of a stronger DN phenotype than the mutant phenotype is a common occurrence in both animals and plants. Compared with the *pxl1* single and *pxl1 pxl2 pxy* triple mutants that only lack the function of known receptors, the Δ Kinase-formed PXL1 shows proper interaction with the CLE peptide and its SERK coreceptors (Fig. 5b-e, k and Fig. S8), but cannot transmit signals downstream. One plausible explanation is that several RLKs, may be more RLKs except for PXL2 and PXY, can perceive the same signal as PXL1 and regulate partially overlapping pathways. The Δ Kinase may sequester and deplete ligands and/or receptor partners of PXL1 that are shared by other RLKs, thereby shutting down the entire pathways, which could function in either a parallel or a convergent manner.

4. The introduction of *pxl1* mutant into DN-PXL1 enhanced the abnormal phenotype, indicating that DN-PXL1 does not completely abolish the function of PXL1. That is due to the expression level of DN-PXL1 is not overwhelmingly higher than that of the WT PXL1. In this case, CLE19+OX was introduced into the DM#8 transgenic plants, in which the relative expression level is 5.55-fold of that of the WT PXL1 in the same double transgenic line. However, DN-PXL1 remains a useful tool for investigating the genetic upstream and downstream relationship between CLE19 and PXL1. As the Δ Kinase form of DN-PXL1 binds to extracellular peptide hormones and SERK coreceptors (Fig. 5b-e, k and Fig. S8), inhibiting normal PXL1 functions through interfering with and shutting down the PXL1 and related RLK pathways that regulate pollen exine development in a partially redundant manner. In this study, we generated CDM double transgenic plants by transforming CLE19-OX into

the DM#8 background, which showed that DN-PXL1 significantly weakened the discontinuous phenotype caused by CLE19-OX on the pollen exine, and presented a pollen wall overfilling phenotype similar to that of the DN-PXL1 and pxl1-related mutants, strongly supporting the functional dependence of CLE19 on PXL1 and its redundant genes.

5. We greatly appreciate the insightful suggestions provided by the referee proposing “the introduction of the loss-of-function of PXL1 (e.g. pxl1-1/-) into the CLE19-OX and examining whether the mutation attenuates the effect of CLE19-OX or not seems more appropriate than the use of DN-PXL1. Simultaneously, evaluating the effect of loss-of-PXL2 or PXY to ask whether the mutations attenuate the abnormalities of CLE19-OX or not also adds molecular and physiological insights into the redundancy and differences of the RLKs.” However, considering the functional redundancy of PXL1, PXL2, and PXY, as well as the weak phenotypes observed in the pxl1 single and pxl1 pxl2 pxy triple mutants compared to the DN-PXL1, we believe that the genetic analysis using CDM offers a more appropriate approach to determine the dependence of CLE19-OX-induced phenotype on PXL1 and its redundant genes. It is worth noting that while the inclusion of the pxl1 mutation in the CDM background may potentially enhance inhibitory effects, the CDM background has already provided a conclusive response to the inquiry at hand.

References:

1. Amaya, E., Musci, T., and Kirschner, M. (1991). Expression of a dominant-negative mutant of the FGF receptor disrupts mesoderm formation in *Xenopus* embryos. *Cell* 66, 257–270.
 2. Hemmati-Brivanlou, A., and Melton, D. (1992). A truncated activin receptor inhibits mesoderm induction and formation of axial structures in *Xenopus* embryos. *Nature* 359, 609–614.
 3. Freeman, M. (1996). Reiterative use of the EGF receptor triggers differentiation of all cell types in the *Drosophila* eye. *Cell* 87, 651–660.
- Shpak ED, Lakeman MB, Torii KU. Dominant-negative receptor uncovers redundancy in the Arabidopsis ERECTA Leucine-rich repeat receptor-like kinase signaling pathway that regulates organ shape. *Plant Cell* 2003, 15:1095-1110.

Figure S5b “Relative expression level PXL1LRR”.
Bracket is disappeared.

Response: Thanks. The bracket has been added.

Figure S5 PXL1FL

I think the description “FL” is confusing. The RT-PCR targeted Kinase Domain, rather than Full Length, the use of KD is preferable.

Response: Thank the referee for this suggestion. We have replaced the “FL” with “KD”.

Figure S5 legend

(c) F12/R2 : typo

Response: Thanks. The “F12/R2” has been revised to “F2/R2”.

Figure 5i/j

Please check the indications of the Y-axis in these graphs. I think these scale lines seem inconsistent, showing in two lines in (upper in i). They make the graphs difficult to understand.

Response: Thanks for pointing this out. The indication of the Y-axis in the upper image of Fig. 5i has been corrected.

Fig S8 anti-MYC bands

According to the Input WB, PXL1KD-MYC and PXL2KD-MYC were detected at similar sizes, conversely, these proteins seem to be detected as different sizes in the IPed WD. The authors are preferred to explain this. If this matter comes from the winding in gel running, the authors are preferred to replace the images with ones that better depict the results.

Response: We appreciate the reviewer for pointing out this issue. The gradual upward slant of the bands from left to right in the previous figure was caused by technical issues, which in turn led to visual misinterpretation. The input protein bands appear continuously from left to right, thus their position differences are not clear; whereas in the IPed sample, several lanes between the PXL1KD-MYC and PXL2KD-MYC bands did not have the same IPed protein, resulting in a visually apparent difference in their positions. In order to present a more objective result, we repeated the experiment and replaced the original images with higher quality ones (Response Fig. 7. and current Fig. S8b).

Response Fig. 7. Fig. S8b of the current manuscript.

The construction detail and plant materials for protoplast assays should be provided.

Response: Thanks. The construction detail and plant materials for protoplast assays have been added to Lines 443-451 in the Method section.

Please re-check the overall statistics included in this manuscript. Especially, the use of p-value is inconsistent through the manuscript, namely the authors sometimes use “p=” and “p<” in a figure e.g. Fig3h/l”. These are very confusing.

Response: Thanks to the referee for pointing this out. In the current manuscript, we have corrected all statistical analysis so that all p-values are represented by “p=”.

Reviewer #1 (Remarks to the Author):

I believe that the authors have responded to the main points that I have raised previously.

Reviewer #2 (Remarks to the Author):

This is the third version of the manuscript that I have reviewed. All points that I have raised have now been addressed.

Reviewer #3 (Remarks to the Author):

The authors have responded appropriately to my concerns, providing additional figures.

I think the current version manuscript is close to the acceptance.

I would raise following minor points;

The authors should reconsider the statistical presentation in fig 2d/3f. It seems that the graphs display the medians of the measurements for each genotype as the longest/thickest bars, rather than the means. However, the authors stated that they used Tukey tests, which is preferable to use mean values to be more clearly shown.

Figure S4

The pattern showing F2/R2 assay was not used.

Point-to-point Responses to Referee #3

The authors should reconsider the statistical presentation in fig 2d/3f. It seems that the graphs display the medians of the measurements for each genotype as the longest/thickest bars, rather than the means. However, the authors stated that they used Tukey tests, which is preferable to use mean values to be more clearly shown.

Response: Thanks. The presentation of the data in figures 2d and 3f have been revised to show means \$\pm\$ SD, as required.

Figure S4

The pattern showing F2/R2 assay was not used.

Response: Thanks for this comment. The pattern showing F2/R2 assay has been added to the current supplementary figure 4b and 4d.